# Learning recurrent dynamics in spiking networks

## Christopher M Kim*, Carson C Chow*

Laboratory of Biological Modeling, National Institute of Diabetes and Digestive and Kidney Diseases, National Institutes of Health, Bethesda, United States

**Abstract** Spiking activity of neurons engaged in learning and performing a task show complex spatiotemporal dynamics. While the output of recurrent network models can learn to perform various tasks, the possible range of recurrent dynamics that emerge after learning remains unknown. Here we show that modifying the recurrent connectivity with a recursive least squares algorithm provides sufficient flexibility for synaptic and spiking rate dynamics of spiking networks to produce a wide range of spatiotemporal activity. We apply the training method to learn arbitrary firing patterns, stabilize irregular spiking activity in a network of excitatory and inhibitory neurons respecting Dale's law, and reproduce the heterogeneous spiking rate patterns of cortical neurons engaged in motor planning and movement. We identify sufficient conditions for successful learning, characterize two types of learning errors, and assess the network capacity. Our findings show that synaptically-coupled recurrent spiking networks possess a vast computational capability that can support the diverse activity patterns in the brain.

DOI: https://doi.org/10.7554/eLife.37124.001

## Introduction

Neuronal populations exhibit diverse patterns of recurrent activity that can be highly irregular or well-structured when learning or performing a behavioral task (*Churchland and Shenoy, 2007*; *Churchland et al., 2012*; *Harvey et al., 2012*; *Pastalkova et al., 2008*; *Li et al., 2015*). An open questions whether learning-induced synaptic rewiring is sufficient to give rise to the wide range of spiking dynamics that encodes and processes information throughout the brain.

It has been shown that a network of recurrently connected neuron models can be trained to perform complex motor and cognitive tasks. In this approach, synaptic connections to a set of outputs are trained to generate a desired population-averaged signal, while the activity of individual neurons within the recurrent network emerges in a self-organized way that harnesses chaotic temporally irregular activity of a network of rate-based neurons (*Sompolinsky et al., 1988*) that is made repeatable through direct feedback from the outputs or through training of the recurrent connections (*Maass et al., 2007*; *Sussillo and Abbott, 2009*). The resulting irregular yet stable dynamics provides a rich reservoir from which complex patterns such as motor commands can be extracted by trained output neurons (*Sussillo and Abbott, 2009*; *Buonomano and Maass, 2009*; *Jaeger and Haas, 2004*), and theoretical studies have shown that the network outputs are able to perform universal computations (*Maass et al., 2007*).

Here, we explore whether there is even a need for a set of output neurons. Instead, each unit in the recurrent network could be considered to be an output and learn target patterns directly while simultaneously serving as a reservoir. *Laje and Buonomano (2013)* showed that individual rate units in a recurrent network can learn to stabilize innate chaotic trajectories that an untrained network naturally generates. The trained trajectories are then utilized to accomplish timing tasks by summing their activities with trained weights. *DePasquale et al. (2018)* obtained a set of target trajectories from a target network driven externally by the desired network output. They showed that training

*For correspondence:
chrismkkim@gmail.com (CMK);
carsonc@niddk.nih.gov (CCC)

**Competing interests:** The authors declare that no competing interests exist.

individual units on such target trajectories and then adjusting the read-out weights yielded better performance than an untrained random recurrent network with a trained feedback loop (i.e. 'traditional' FORCE learning). *Rajan et al., 2016* trained a small fraction of synaptic connections in a randomly connected rate network to produce sequential activity derived from cortical neurons engaged in decision making tasks.

Although these studies demonstrate that units within a rate-based network can learn recurrent dynamics defined by specific forms of target functions, the possible repertoire of the recurrent activity that a recurrent network can learn has not been extensively explored. Moreover, extending this idea to spiking networks, where neurons communicate with time dependent spike induced synapses, poses an additional challenge because it is difficult to coordinate the spiking dynamics of many neurons, especially, if spike times are variable as in a balanced network (*London et al., 2010*). Some success has been achieved by training spiking networks directly with a feedback loop (*Nicola and Clopath, 2017*) or using a rate-based network as an intermediate step (*DePasquale et al., 2016*; *Thalmeier et al., 2016*). A different top-down approach is to build networks that emit spikes optimally to correct the discrepancy between the actual and desired network outputs (*Boerlin et al., 2013*; *Denève and Machens, 2016*). This optimal coding strategy in a tightly balanced network can be learned with a local plasticity rule (*Brendel et al., 2017*) and is able to generate arbitrary network output at the spike level (*Bourdoukan and Deneve, 2015*; *Denève et al., 2017*).

We show that a network of spiking neurons is capable of supporting arbitrarily complex coarse-grained recurrent dynamics provided the spatiotemporal patterns of the recurrent activity are diverse, the synaptic dynamics are fast, and the number of neurons in the network is large. We give a theoretical basis for how a network can learn and show various examples, which include stabilizing strong chaotic rate fluctuations in a network of excitatory and inhibitory neurons that respects Dale's law and constructing a recurrent network that reproduces the spiking rate patterns of a large number of cortical neurons involved in motor planning and movement. Our study suggests that individual neurons in a recurrent network have the capability to support near universal dynamics.

## Results

### Spiking networks can learn complex recurrent dynamics

We considered a network of $N$ quadratic integrate-and-fire neurons that are recurrently connected with spike-activated synapses weighted by a connectivity matrix $W$. We show below that our results do not depend on the spiking mechanism. We focused on two measures of coarse-grained time-dependent neuron activity: (1) the synaptic drive $u_i(t)$ to neuron $i$ which is given by the $W$-weighted sum of low-pass filtered incoming spike trains, and (2) the time-averaged spiking rate $R_i(t)$ of neuron $i$. The goal was to find a weight matrix $W$ that can autonomously generate desired recurrent target dynamics when the network of spiking neurons connected by $W$ is stimulated briefly with an external stimulus (*Figure 1a*). The target dynamics were defined by a set of functions $f_1(t), f_2(t), ..., f_N(t)$ on a time interval $[0, T]$. Learning of the recurrent connectivity $W$ was considered successful if $u_i(t)$ or $R_i(t)$ evoked by the stimulus matches the target functions $f_i(t)$ over the time interval $[0, T]$ for all neurons $i = 1, 2, ..., N$.

Previous studies have shown that recurrently connected rate units can learn specific forms of activity patterns, such as chaotic trajectories that the initial network could already generate (*Laje and Buonomano, 2013*), trajectories from a target network (*DePasquale et al., 2018*), and sequential activity derived from imaging data (*Rajan et al., 2016*). Our study expanded these results in two ways: first, we trained the recurrent dynamics of spiking networks, and, second, we showed that the repertoire of recurrent dynamics that can be encoded is vast. The primary goal of our paper was to investigate the computational capability of spiking networks to generate arbitrary recurrent dynamics, therefore we neither trained the network outputs (*Sussillo and Abbott, 2009*; *Sussillo et al., 2015*; *Nicola and Clopath, 2017*) nor constrained the target signals to those designed for performing specific computations (*DePasquale et al., 2018*). We focused on training the recurrent activity as in the work of *Laje and Buonomano (2013)* (without the read-outs) and *Rajan et al., 2016*, and considered arbitrary target functions. To train the activity of individual neurons within a spiking network, we extended the Recursive Least Squares (RLS) algorithm developed by Laje and Buonomano in rate-based networks (*Laje and Buonomano, 2013*). The algorithm was

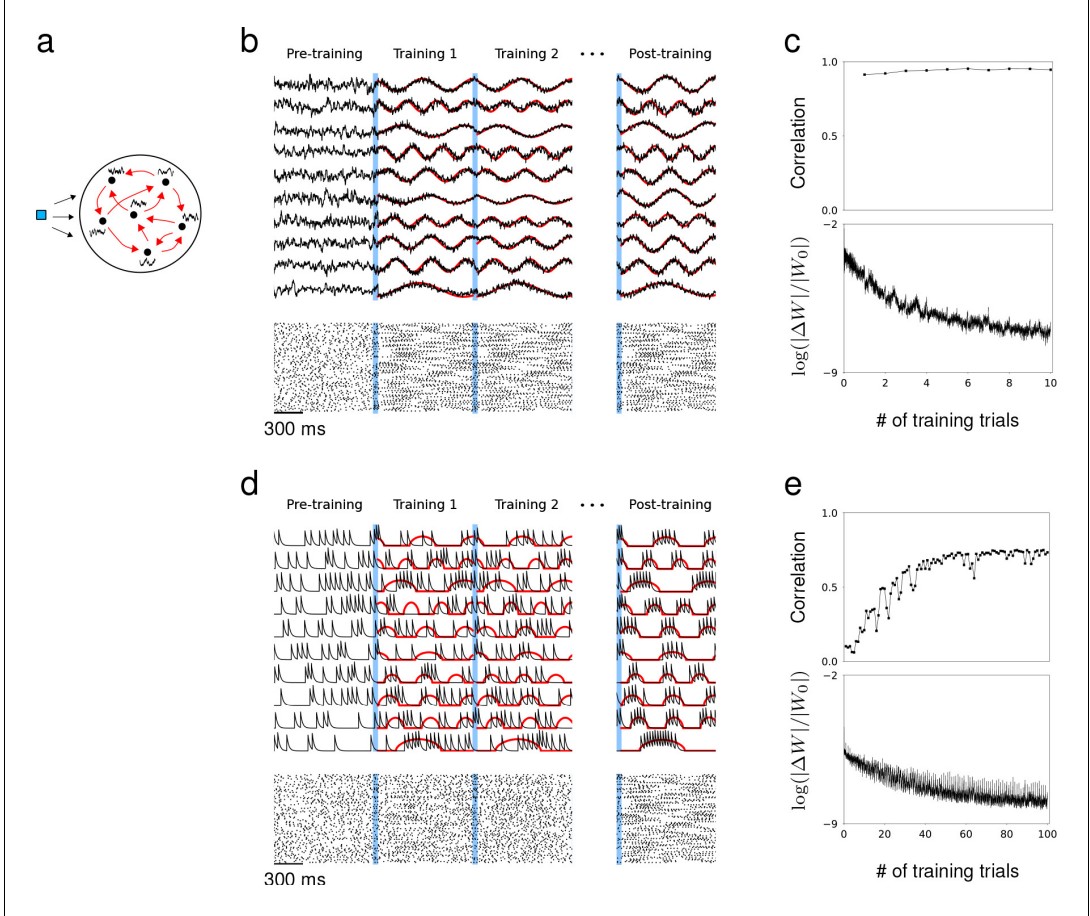

**Figure 1.** Synaptic drive and spiking rate of neurons in a recurrent network can learn complex patterns. (**a**) Schematic of network training. Blue square represents the external stimulus that elicits the desired response. Black curves represent target output for each neuron. Red arrows represent recurrent connectivity that is trained to produce desired target patterns. (**b**) Synaptic drive of 10 sample neurons before, during and after training. Pre-training is followed by multiple training trials. An external stimulus (blue) is applied prior to training for 100 ms. Synaptic drive (black) is trained to follow the target (red). If the training is successful, the same external stimulus can elicit the desired response. Bottom shows the spike rater of 100 neurons. (**c**) Top, The Pearson correlation between the actual synaptic drive and the target output during training trials. Bottom, The matrix (Fresenius) norm of changes in recurrent connectivity normalized to initial connectivity during training. (**d**) Filtered spike train of 10 neurons before, during and after training. As in (**b**), external stimulus (blue) is applied immediately before training trials. Filtered spike train (black) learns to follow the target spiking rate (red) with large errors during the early trials. Applying the stimulus to a successfully trained network elicits the desired spiking rate patterns in every neuron. (**e**) Top, Same as in (**c**) but measures the correlation between filtered spike trains and target outputs. Bottom, Same as in (**c**).

DOI: https://doi.org/10.7554/eLife.37124.002

The following figure supplements are available for figure 1:

**Figure supplement 1.** Learning arbitrarily complex target patterns in a network of rate-based neurons.

DOI: https://doi.org/10.7554/eLife.37124.003

**Figure supplement 2.** Training a network that has no initial connections.

DOI: https://doi.org/10.7554/eLife.37124.004

based on the FORCE algorithm (*Haykin, 1996*; *Sussillo and Abbott, 2009*), originally developed to train the network outputs by minimizing a quadratic cost function between the activity measure and the target together with a quadratic regularization term (see Materials and methods, 'Training recurrent dynamics'). Example code that trains a network of quadratic integrate-and-fire neurons is available at https://github.com/chrismkkim/SpikeLearning (*Kim and Chow, 2018*; copy archived at https://github.com/elifesciences-publications/SpikeLearning).

As a first example, we trained the network to produce synaptic drive patterns that matched a set of sine functions with random frequencies and the spiking rate to match the positive part of the same sine functions. The initial connectivity matrix had connection probability $p = 0.3$ and the

coupling strength was drawn from a Normal distribution with mean $0$ and standard deviation $\sigma$. Prior to training, the synaptic drive fluctuated irregularly, but as soon as the RLS algorithm was instantiated, the synaptic drives followed the target with small error; rapid changes in $W$ quickly adjusted the recurrent dynamics towards the target (*Sussillo and Abbott, 2009*) (*Figure 1b,c*). As a result, the population spike trains exhibited reproducible patterns across training trials. A brief stimulus preceded each training session to reset the network to a specific state. If the training was successful, the trained response could be elicited whenever the same stimulus was applied regardless of the network state. We were able to train a network of rate-based neurons to learn arbitrarily complex target patterns using the same learning scheme (*Figure 1—figure supplement 1*).

Training the spiking rate was more challenging than training the synaptic drive because small changes in recurrent connectivity did not immediately affect the spiking activity if the effect was below the spike-threshold. Therefore, the spike trains may not follow the desired spiking rate pattern during the early stage of training, and the population spike trains no longer appeared similar across training trials (*Figure 1d*). This was also reflected in relatively small changes in recurrent connectivity and the substantially larger number of training runs required to produce desired spiking patterns (*Figure 1e*). However, by only applying the training when the total input to a neuron is suprathreshold, the spiking rate could be trained to reproduce the target patterns. The correlation between the actual filtered spike trains and the target spiking rate increased gradually as the training progressed.

Previous work that trained the network read-out had proposed that the initial recurrent network needed to be at the 'edge of chaos' to learn successfully (*Bertschinger and Natschläger, 2004*; *Sussillo and Abbott, 2009*; *Abbott et al., 2016*; *Thalmeier et al., 2016*; *Nicola and Clopath, 2017*). However, we found that the recurrent connectivity could learn to produce the desired recurrent dynamics regardless of the initial network dynamics and connectivity. Even when the initial network had no synaptic connections, the brief stimulus preceding the training session was sufficient to build a fully functioning recurrent connectivity that captured the target dynamics. The RLS algorithm could grow new synapses or tune existing ones as long as some of the neurons became active after the initial stimulus (*Figure 1—figure supplement 2*).

Learning was not limited to one set of targets; the same network was able to learn multiple sets of targets. We trained the network to follow two independent sets of targets, where each target function was a sine function with random frequency. Every neuron in the network learned both activity patterns after training, and, when stimulated with the appropriate cue, the network recapitulated the specified trained pattern of recurrent dynamics, regardless of initial activity. The synaptic drive and the spiking rate were both able to learn multiple target patterns (*Figure 2*).

## Learning arbitrary patterns of activity

Next, we considered targets generated from various families of functions: complex periodic functions, chaotic trajectories, and Ornstein-Hollenbeck (OU) noise. We randomly selected $N$ different

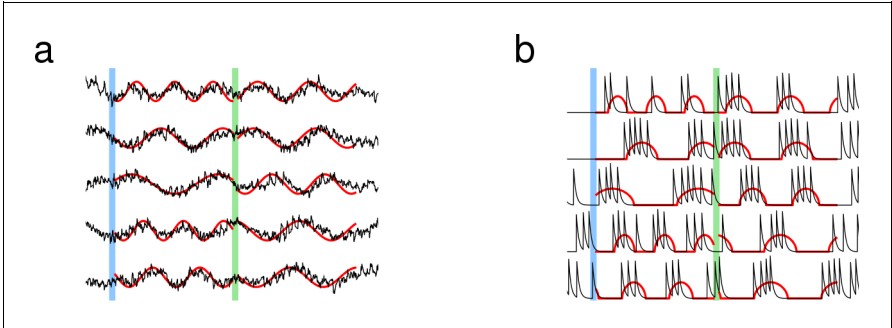

**Figure 2.** Learning multiple target patterns. (**a**) The synaptic drive of neurons learns two different target outputs. Blue stimulus evokes the first set of target outputs (red) and the green stimulus evokes the second set of target outputs (red). (**b**) The spiking rate of individual neurons learns two different target outputs.

DOI: https://doi.org/10.7554/eLife.37124.005

target patterns from one of the families to create a set of heterogeneous targets, and trained the synaptic drive of a network consisting of $N$ neurons to learn the target dynamics. These examples demonstrated that recurrent activity patterns that a spiking network can generate is not limited to specific forms of patterns considered in previous studies (*Laje and Buonomano, 2013*; *Rajan et al., 2016*; *DePasquale et al., 2018*), but can be arbitrary functions. The successful learning suggested that single neurons embedded in a spiking network have the capability to perform universal computations.

As we will show more rigorously in the next section, we identified two sufficient conditions on the dynamical state and spatiotemporal structure of target dynamics that ensure a wide repertoire of recurrent dynamics can be learned. The first is a 'quasi-static' condition that stipulates that the dynamical time scale of target patterns must be slow enough compared to the synaptic time scale and average spiking rate. The second is a 'heterogeneity' condition that requires the spatiotemporal structure of target patterns to be diverse enough. The target patterns considered in *Figure 3* had slow temporal dynamics in comparison to the synaptic time constant ($\tau_s = 20$ ms) and the patterns were selected randomly to promote diverse structure. After training each neuron's synaptic drive to produce the respective target pattern, the synaptic drive of every neuron in the network followed its target.

To verify the quasi-static condition, we compared the actual to a quasi-static approximation of the spiking rate and synaptic drive. The spiking rates of neurons were approximated using the current-to-rate transfer function with time-dependent synaptic input, and the synaptic drive was approximated by a weighted sum of the presynaptic neurons' spiking rates. We elicited the trained patterns over multiple trials starting at random initial conditions to calculate the trial-averaged spiking rates. The quasi-static approximations of the synaptic drive and spiking rate closely matched the actual synaptic drive (*Figure 3a*) and trial-averaged spiking rates (*Figure 3b*).

To examine how the heterogeneity of target patterns may facilitate learning, we created sets of target patterns where the fraction of randomly generated targets was varied systematically. For non-random targets, we used the same target pattern repeatedly. Networks trained to learn target patterns with strong heterogeneity showed that a network is able to encode target patterns with high accuracy if there is a large fraction of random targets (*Figure 3c*). Networks that were trained on too many repeated target patterns failed to learn. Beyond a certain fraction of random patterns, including additional patterns did not improve the performance, suggesting that the set of basis functions was over-complete. We probed the stability of over-complete networks under neuron loss by eliminating all the synaptic connections from a fraction of the neurons. A network was first trained to learn target outputs where all the patterns were selected randomly (i.e. fraction of random targets equals 1) tonsure that the target patterns form a set of redundant basis functions. Then, we elicited the trained patterns after removing a fraction of neurons from the network, which entails eliminating all the synaptic connections from the lost neurons. A trained network with 5% neuron loss was able to generate the trained patterns perfectly, 10% neuron loss resulted in a mild degradation of network response, and trained patterns completely disappeared after 40% neuron loss (*Figure 3d*).

The target dynamics considered in *Figure 3* had population spiking rates of 9.1 Hz (periodic), 7.2 Hz (chaotic) and 12.1 Hz (OU) within the training window. To examine how population activity may influence learning, we trained networks to learn target patterns whose average amplitude was reduced gradually across target sets. The networks were able to learn when the population spiking rate of the target dynamics was as low as 1.5 Hz. However, the performance deteriorated as the population spiking rate decreased further (*Figure 3—figure supplement 1*). To demonstrate that learning does not depend on the spiking mechanism, we trained the synaptic drive of spiking networks using different neuron models. A network of leaky integrate-and-fire neurons, as well as a network of Izhikevich neurons whose neuron parameters were tuned to have five different firing patterns, successfully learned complex synaptic drive patterns (*Figure 3—figure supplement 2*).

## Stabilizing rate fluctuations in a network respecting Dale's law

A random network with balanced excitation and inhibition is a canonical model for a cortical circuit that produces asynchronous single unit activity (*Sompolinsky et al., 1988*; *van Vreeswijk et al., 1996*; *Renart et al., 2010*; *Ostojic, 2014*; *Rosenbaum et al., 2017*). The chaotic activity of balanced rate models (*Sompolinsky et al., 1988*) has been harnessed to accomplish complex tasks by

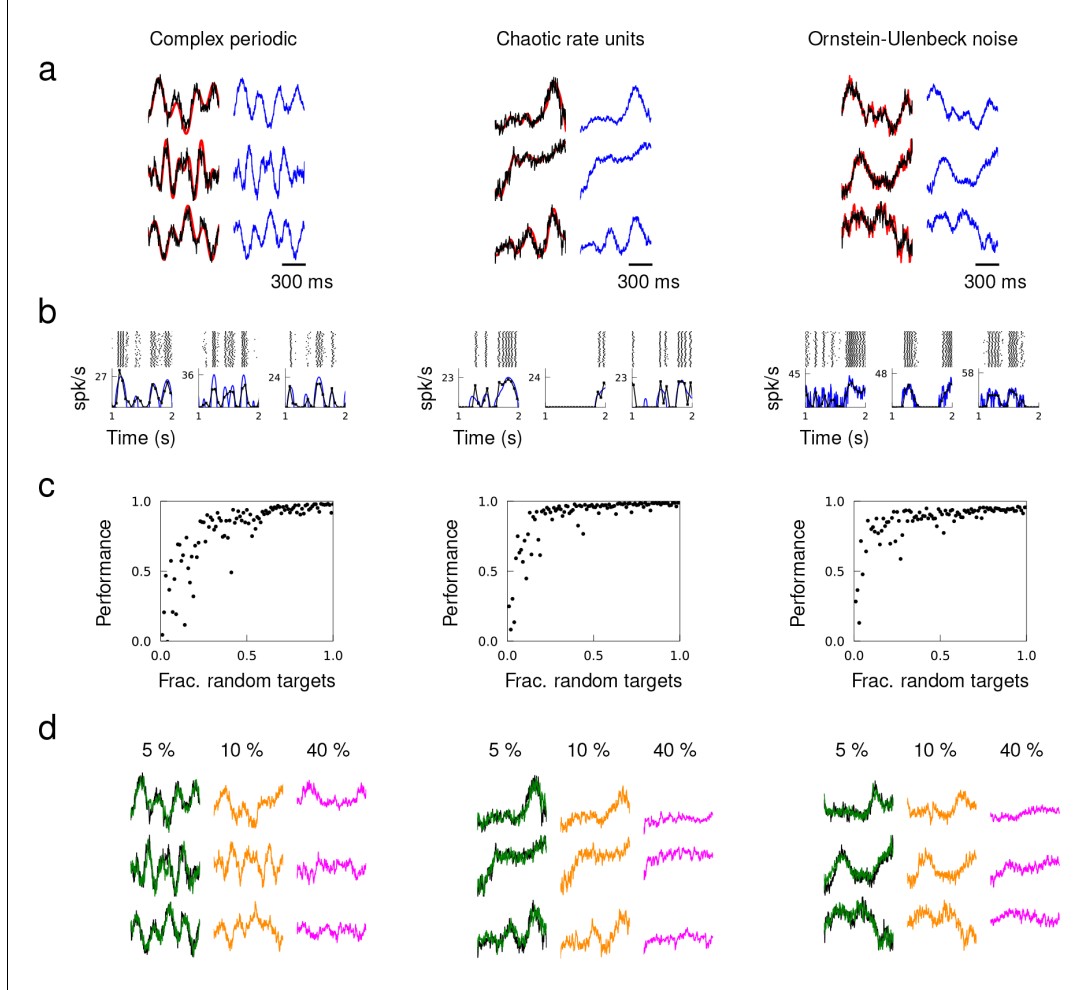

**Figure 3.** Quasi-static and heterogeneous patterns can be learned. Example target patterns include complex periodic functions (product of sines with random frequencies), chaotic rate units (obtained from a random network of rate units), and OU noise (obtained by low-pass filtering white noise with time constant 100 ms). (a) Target patterns (red) overlaid with actual synaptic drive (black) of a trained network. Quasi-static prediction (*Equation 1*) of synaptic drive (blue). (b) Spike trains of trained neurons elicited multiple trials, trial-averaged spiking rate calculated by the average number of spikes in 50 ms time bins (black), and predicted spiking rate (blue). (c) Performance of trained network as a function of the fraction of randomly selected targets. (d) Network response from a trained network after removing all the synaptic connections from 5%, 10% and 40% of randomly selected neurons in the network.

DOI: https://doi.org/10.7554/eLife.37124.006

The following figure supplements are available for figure 3:

**Figure supplement 1.** Learning target patterns with low-population spiking rate.

DOI: https://doi.org/10.7554/eLife.37124.007

**Figure supplement 2.** Learning recurrent dynamics with leaky integrate-and-fire and Izhikevich neuron models.

DOI: https://doi.org/10.7554/eLife.37124.008

**Figure supplement 3.** Synaptic drive of a network of neurons is trained to learn an identical sine wave while external noise generated independently from OU process is injected to individual neurons.

DOI: https://doi.org/10.7554/eLife.37124.009

including a feedback loop (*Sussillo and Abbott, 2009*), stabilizing chaotic trajectories (*Laje and Buonomano, 2013*) or introducing low-rank structure to the connectivity matrix (*Mastrogiuseppe and Ostojic, 2017*). Balanced spiking networks have been shown to possess similar capabilities (*Thalmeier et al., 2016*; *DePasquale et al., 2016*; *Abbott et al., 2016*; *Nicola and Clopath, 2017*; *Denève and Machens, 2016*), but it is unknown if it is possible to stabilize the heterogeneous fluctuations of the spiking rate in the strong coupling regime (*Ostojic, 2014*). Here, we extended the work of *Laje and Buonomano (2013)* to spiking networks and showed that strongly

fluctuating single neuron activities can be turned into dynamic attractors by adjusting the recurrent connectivity.

We considered a network of randomly connected excitatory and inhibitory neurons that respected Dale's Law. Prior to training, the synaptic and spiking activity of individual neurons showed large variations across trials because small discrepancies in the initial network state led to rapid divergence of network dynamics. When simulated with two different initial conditions, the synaptic drive to neurons deviated strongly from each other (*Figure 4a*), and the spiking activity of single neurons was uncorrelated across trials and the trial-averaged spiking rate had little temporal structure (*Figure 4b*). The network activity was also sensitive to small perturbation; the microstate of two identically prepared networks diverged rapidly if one spike was deleted from one of the

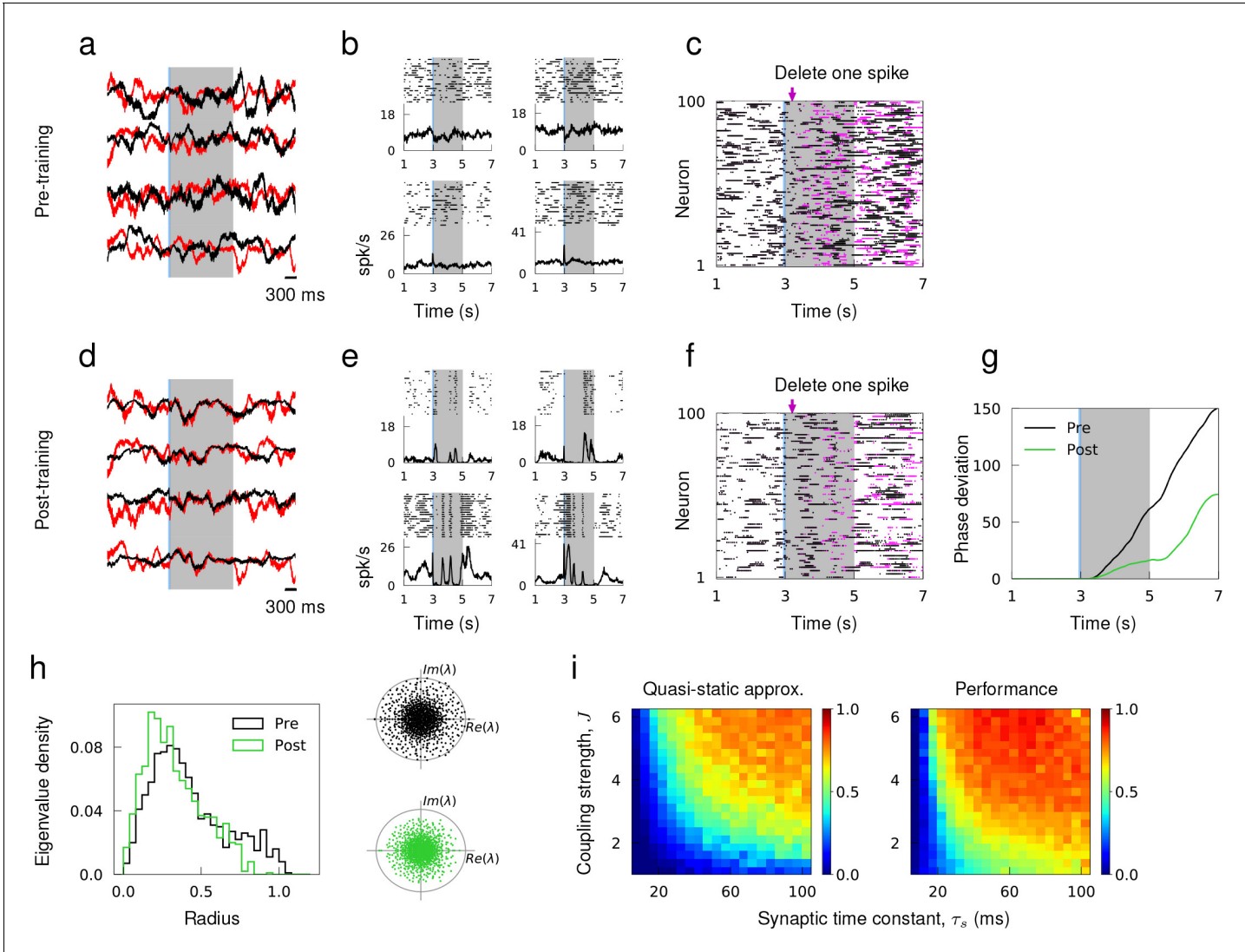

**Figure 4.** Learning innate activity in a network of excitatory and inhibitory neurons that respects Dale's Law. (a) Synaptic drive of sample neurons starting at random initial conditions in response to external stimulus prior to training. (b) Spike raster of sample neurons evoked by the same stimulus over multiple trials with random initial conditions. (c) Single spike perturbation of an untrained network. (d)-(f) Synaptic drive, multi-trial spiking response and single spike perturbation in a trained network. (g) The average phase deviation of theta neurons due to single spike perturbation. (h) Left, distribution of eigenvalues of the recurrent connectivity before and after training as a function their absolution values. Right, Eigenvalue spectrum of the recurrent connectivity; gray circle has unit radius. (i) The accuracy of quasi-static approximation in untrained networks and the performance of trained networks as a function of coupling strength J and synaptic time constant $\tau_s$. Color bar shows the Pearson correlation between predicted and actual synaptic drive in untrained networks (left) and innate and actual synaptic drive in trained networks (right).

DOI: https://doi.org/10.7554/eLife.37124.010

networks (*Figure 4c*). It has been previously questioned as to whether the chaotic nature of an excitatory-inhibitory network could be utilized to perform reliable computations (*London et al., 2010*; *Monteforte and Wolf, 2012*).

As in *Laje and Buonomano (2013)*, we sought to tame the chaotic trajectories of single neuron activities when the coupling strength is strong enough to induce large and irregular spiking rate fluctuations in time and across neurons (*Ostojic, 2014*). We initiated the untrained network with random initial conditions to harvest innate synaptic activity, that is a set of synaptic trajectories that the network already knows how to generate. Then, the recurrent connectivity was trained so that the synaptic drive of every neuron in the network follows the innate pattern when stimulated with an external stimulus. To respect Dale's Law, the RLS learning rule was modified such that it did not update synaptic connections if there were changes in their signs.

After training, the synaptic drive to every neuron in the network was able to track the innate trajectories in response to the external stimulus within the trained window and diverged from the target pattern outside the trained window (*Figure 4d*). When the trained network was stimulated to evoke the target patterns, the trial-averaged spiking rate developed a temporal structure that was not present in the untrained network (*Figure 4e*). To verify the reliability of learned spiking patterns, we simulated the trained network twice with identical initial conditions but deleted one spike 200 ms after evoking the trained response from one of the simulations. Within the trained window, the relative deviation of the microstate was markedly small in comparison to the deviation observed in the untrained network. Outside the trained window, however, two networks diverged rapidly again, which demonstrated that training the recurrent connectivity created an attracting flux tube around what used to be chaotic spike sequences (*Monteforte and Wolf, 2012*) (*Figure 4f,g*). Analyzing the eigenvalue spectrum of the recurrent connectivity revealed that the distribution of eigenvalues shifts towards zero and the spectral radius decreased as a result of training, which is consistent with the more stable network dynamics found in trained networks (*Figure 4h*).

To demonstrate that learning the innate trajectories works well when an excitatory-inhibitory network satisfies the quasi-static condition, we scanned the coupling strength $J$ (see Materials and methods, 'Training recurrent dynamics' for the definition) and synaptic time constant $\tau_s$ over a wide range and evaluated the accuracy of the quasi-static approximation in untrained networks. We find that increasing either $J$ or $\tau_s$ promoted strong fluctuations in spiking rates (*Ostojic, 2014*; *Harish and Hansel, 2015*), hence improving the quasi-static approximation (*Figure 4i*). Learning performance was correlated with adherence to the quasi-static approximation, resulting in better performance for strong coupling and long synaptic time constants.

## Generating an ensemble of in vivo spiking patterns

We next investigated if the training method applied to actual spike recordings of a large number of neurons. In a previous study, a network of rate units was trained to match sequential activity imaged from posterior parietal cortex as a possible mechanism for short-term memory (*Harvey et al., 2012*; *Rajan et al., 2016*). Here, we aimed to construct recurrent spiking networks that captured heterogeneous spiking activity of cortical neurons involved in motor planning and movement (*Churchland and Shenoy, 2007*; *Churchland et al., 2012*; *Li et al., 2015*).

The in vivo spiking data was obtained from the publicly available data of *Li et al. (2015)*, where they recorded the spike trains of a large number of neurons from the anterior lateral motor cortex of mice engaged in planning and executing directed licking over multiple trials. We compiled the trial-average spiking rate of $N_{\mathrm{cor}} = 227$ cortical neurons from their data set (*Li et al., 2014*), and trained a recurrent network model to reproduce the spiking rate patterns of all the $N_{cor}$ neurons autonomously in response to a brief external stimulus. We only trained the recurrent connectivity and did not alter single neuron dynamics or external inputs.

First, we tested if a recurrent network of size $N_{\mathrm{cor}}$ was able to generate the spiking rate patterns of the same number of cortical neurons. This network model assumed that the spiking patterns of $N_{\mathrm{cor}}$ cortical neurons could be self-generated within a recurrent network. After training, the spiking rate of neuron models captured the overall trend of the spiking rate, but not the rapid changes that may be pertinent to the short term memory and motor response (*Figure 5b*). We hypothesized that the discrepancy may be attributed to other sources of input to the neurons not included in the model, such as recurrent input from other neurons in the local population or input from other areas

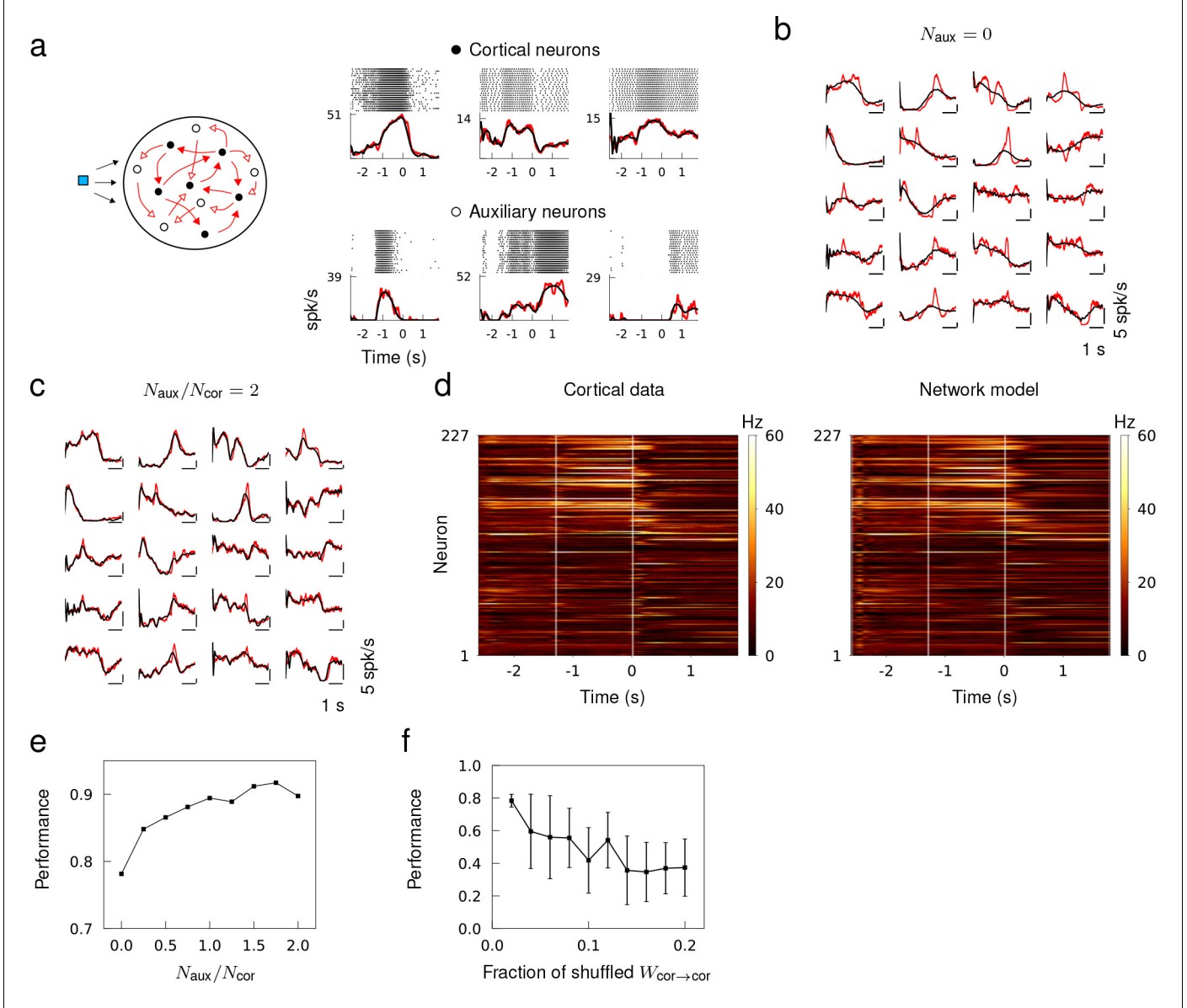

**Figure 5.** Generating in vivo spiking activity in a subnetwork of a recurrent network. (**a**) Network schematic showing cortical (black) and auxiliary (white) neuron models trained to follow the spiking rate patterns of cortical neurons and target patterns derived from OU noise, respectively. Multi-trial spike sequences of sample cortical and auxiliary neurons in a successfully trained network. (**b**) Trial-averaged spiking rate of cortical neurons (red) and neuron models (black) when no auxiliary neurons are included. (**c**) Trial-averaged spiking rate of cortical and auxiliary neuron models when $N_{aux} = N_{aux} = 2$. (**c**) Spiking rate of all the cortical neurons from the data (left) and the recurrent network model (right) trained with $N_{aux} = N_{cor} = 2$. (**e**) The fit to cortical dynamics improves as the number of auxiliary neurons increases. (**f**) Random shuffling of synaptic connections between cortical neuron models degrades the fit to cortical data. Error bars show the standard deviation of results from 10 trials.

DOI: https://doi.org/10.7554/eLife.37124.011

of the brain, or the neuron dynamics that cannot be captured by our neuron model. We thus sought to improve the performance by adding $N_{aux}$ auxiliary neurons to the recurrent network to mimic the spiking activity of unobserved neurons in the local population, and trained the recurrent connectivity of a network of size $N_{cor} = N_{aux} = 2$ (**Figure 5a**). The auxiliary neurons were trained to follow spiking rate patterns obtained from an OU process and provided heterogeneity to the overall population activity patterns. When $N_{aux}/N_{cor} \geq 2$, the spiking patterns of neuron models accurately fit that of cortical neurons (**Figure 5c**), and the population activity of all $N_{cor}$ cortical neurons was well captured

by the network model (*Figure 5d*). The fit to cortical activity improved gradually as a function of the fraction of auxiliary neurons in the network due to increased heterogeneity in the target patterns (*Figure 5e*)

To verify that the cortical neurons in the network model were not simply driven by the feed forward inputs from the auxiliary neurons, we randomly shuffled a fraction of recurrent connections between cortical neurons after a successful training. The fit to cortical data deteriorated as the fraction of shuffled synaptic connections between cortical neurons was increased, which confirmed that the recurrent connections between the cortical neurons played a role in generating the spiking patterns (*Figure 5f*).

## Sufficient conditions for learning

We can quantify the sufficient conditions the target patterns need to satisfy in order to be successfully encoded in a network. The first condition is that the dynamical time scale of both neurons and synapses must be sufficiently fast compared to the target patterns such that targets can be considered constant (quasi-static) on a short time interval. In terms of network dynamics, the quasi-static condition implies that the synaptic and neuron dynamics operate as if in a stationary state even though the stationary values change as the network activity evolves in time. In this quasi-static state, we can use a mean field description of the spiking dynamics to derive a self-consistent equation that captures the time-dependent synaptic and spiking activity of neurons (*Buice and Chow, 2013*; *Ostojic, 2014*; *Brunel, 2000*) (see Materials and methods, 'Mean field description of the quasi-static dynamics'). Under the quasi-static approximation, the synaptic drive satisfies

$$U_i(t) = \sum_{j=1}^{N} W_{ij}\phi\big(U_j(t) + I_j\big), \tag{1}$$

and the spiking rate $R_i = \phi(U_i + I_i)$ satisfies

$$R_i(t) = \phi\left(\sum_{j=1}^{N} W_{ij}R_j(t)\right), \tag{2}$$

where $\phi$ is the current-to-rate transfer (i.e. gain) function and $I_i$ is a constant external input.

The advantage of operating in a quasi-static state is that both measures of network activity become conducive to learning new patterns. First, *Equation 1* is closed in terms of $U$, which implies that training the synaptic drive is equivalent to training a rate-based network. Second, the RLS algorithm can efficiently optimize the recurrent connectivity $W$, thanks to the linearity of *Equation 1* in $W$, while the synaptic drive closely follows the target patterns as shown in *Figure 1b*. The spiking rate also provides a closed description of the network activity, as described in *Equation 2*. However, due to nonlinearity in $W$, it learns only when the total input to a neuron is supra-threshold, that is the gradient of $\phi$ must be positive. For this reason, the learning error cannot be controlled as tightly as the synaptic drive and requires additional trials for successful learning as shown in *Figure 1d*.

The second condition requires the target patterns to be sufficiently heterogeneous in time and across neurons. Such complexity allows the ensemble of spiking activity to have a rich spatiotemporal structure to generate the desired activity patterns of every neuron within the network. In the perspective of 'reservoir computing' (*Maass et al., 2002*; *Jaeger and Haas, 2004*; *Sussillo and Abbott, 2009*), every neuron in a recurrent network is considered to be a read-out, and, at the same time, it is part of the reservoir that is collectively used to produce desired patterns in single neurons. The heterogeneity condition is equivalent to having a set of complete (or over-complete) basis functions, that is $\phi(U_j + I_j), j = 1, ..., N$ in *Equation 1* and $R_j, j = 1, ..., N$ in *Equation 2*, to generate the target patterns, that is the left hand side of *Equations 1 and 2*. The two conditions are not necessarily independent. Heterogeneous targets also foster asynchronous spiking activity that support quasi-static dynamics.

We can illustrate the necessity of heterogeneous target functions with a simple argument. Successful learning is achieved for the synaptic drive when *Equation 1* is satisfied. If we discretize time into $P$ 'quasi-static' bins then we can consider the target $U_i(t)$ as a $N \times P$ matrix that satisfies the system of equations expressed in matrix form as $U = WV$, where $V \equiv \phi(U + I)$ is an $N \times P$ matrix. Since the elements of $W$ are the unknowns, it is convenient to consider the transpose of the matrix

equation, $U^T = V^T W^T$. Solving for $W^T$ is equivalent to finding $\mathbf{w}_i$ in $\mathbf{u}_i = V^T \mathbf{w}_i$ for $i = 1, ..., N$, where $\mathbf{u}_i$ is a vector in $P$-dimensional Euclidean space $\mathbb{R}^P$ denoting the $i^{\text{th}}$ column of $U^T$ (the synaptic drive of neuron $i$) and $\mathbf{w}_i$ is an $N$-dimensional vector denoting the $i^{\text{th}}$ column of $W^T$ (the incoming synaptic connections to neuron $i$). We also denote the column vectors of $V^T$ in $\mathbb{R}^P$ by $\mathbf{v}_1, ..., \mathbf{v}_N$ (the firing rate patterns of neurons induced by the target functions). For each $i$, the system of equations consists of $P$ equations and $N$ unknowns.

In general, the system of equations is solvable if all target functions $\mathbf{u}_i, i = 1, ..., N$ lie in the subspace spanned by $\mathbf{v}_1, ..., \mathbf{v}_N$. This is equivalent to stating that the target functions can be self-consistently generated by the firing rate patterns induced by the target functions. We define target functions to be sufficiently heterogeneous if $\text{rank}(V)$ is maximal and show that this is a sufficient condition for solutions to exist. Since the span of $\mathbf{v}_1, ..., \mathbf{v}_N$ encompasses the largest possible subspace in $\mathbb{R}^P$ if $\text{rank}(V)$ is maximal, it is justified as a mathematical definition of sufficiently heterogeneous. In particular, if $N \geq P$ and $\text{rank}(V)$ is maximal, we have $\dim \text{span}\{\mathbf{v}_1, ..., \mathbf{v}_N\} = P$, which implies that the set of firing rate vectors $\mathbf{v}_1, ..., \mathbf{v}_N$ fully span $\mathbb{R}^P$, of which the target vectors $\mathbf{u}_i$ are elements; in other words, $\mathbf{v}_1, ..., \mathbf{v}_N$ forms an (over-)complete set of basis functions of $\mathbb{R}^P$. On the other hand, if $N < P$ and $\text{rank}(V)$ is maximal, we have $\dim \text{span}\{\mathbf{v}_1, ..., \mathbf{v}_N\} = N$, which implies linearly independent $\mathbf{v}_1, ..., \mathbf{v}_N$ can only span an $N$-dimensional subspace of $\mathbb{R}^P$, but such subspace still attains the largest possible dimension.

Now we consider the solvability of $\mathbf{u}_i = V^T \mathbf{w}_i$ when $\text{rank}(V)$ is maximal. For $N \geq P$, the set of vectors $\mathbf{v}_1, ..., \mathbf{v}_N$ fully span $\mathbb{R}^P$, or equivalently we can state that there are more unknowns ($N$) than independent equations ($P$), in which case the equation can always be satisfied and learning the pattern is possible. If $N$ is strictly larger than $P$ then a regularization term is required for the algorithm to converge to a specific solution out of the many possible solutions, the number of which decreases as $P$ approaches $N$. For $N < P$, on the other hand, $\mathbf{v}_1, ..., \mathbf{v}_N$ spans an $N$-dimensional subspace of $\mathbb{R}^P$, or equivalently there will be more equations than unknowns and perfect learning is not possible. However, since $\text{rank}(V)$ is maximal, there is an approximate regression solution of the form $W = UV^T(VV^T)^{-1}$, where the inverse of $VV^T$ exists since the set of vectors $\mathbf{v}_1, ..., \mathbf{v}_N$ is linearly independent.

When $\text{rank}(V)$ is not maximal, successful learning is still possible as long as all $\mathbf{u}_i, i = 1, ..., N$ lie close to the subspace spanned by $\mathbf{v}_1, ..., \mathbf{v}_N$. However, the success depends on the specific choice of target functions, because the dimension of the subspace spanned by $\mathbf{v}_1, ..., \mathbf{v}_N$ is strictly less than $P$, so whether the rows of $U$ are contained in or close to this subspace is determined by the geometry of the subspace. This shows why increasing pattern heterogeneity, which makes the columns of $V^T$ more independent and the rank higher, is beneficial for learning. Conversely, as a larger number of neurons is trained on the same target, as considered in *Figure 3c*, it becomes increasingly difficult to develop the target pattern $\mathbf{u}_i$ with the limited set of basis functions $\mathbf{v}_1, ..., \mathbf{v}_N$.

This argument also shows why learning capability declines as $P$ increases, with a steep decline for $P > N$. If we ascribe a quasi-static bin to some fraction of the pattern correlation time then $P$ will scale with the length of the pattern temporal length. In this way, we can intuitively visualize the temporal storage capacity demonstrated below in Figure 7 through simulations.

We note that although *Equations 1 and 2* describe the dynamical state in which learning works well, merely finding $W$ that satisfies one of the equations does not guarantee that a spiking network with recurrent connectivity $W$ will produce the target dynamics in a stable manner. The recurrent connectivity $W$ needs to be trained iteratively as the network dynamics unfold in time to ensure that the target dynamics is generated in a stable manner (*Sussillo and Abbott, 2009*). There are three aspects of the training scheme that promote stable dynamics around the target trajectories. First, the stimulus at the onset of the learning window is applied at random times so it only sets the initial network states close to each other but with some random deviations. Training with initial conditions sampled from a small region in the state space forces the trained network to be robust to the choice of initial condition, and the target dynamics can be evoked reliably. Second, various network states around the target trajectories are explored while $W$ is learning the desired dynamics. In-between the time points when $W$ is updated, the network states evolve freely with no constraints and can thus diverge from the desired trajectory. This allows the network to visit different network states in the neighborhood of the target trajectories during training, and the trained network becomes resistant to relatively small perturbations from the target trajectories. Third, the synaptic update rule is

designed to reduce the error between the target and the ongoing network activity each time $W$ is updated. Thus, the sequential nature of the training procedure automatically induces stable dynamics by contracting trajectories toward the target throughout the entire path. In sum, robustness to initial conditions and network states around the target trajectories, together with the contractive property of the learning scheme, allow the trained network to generate the target dynamics in a stable manner.

## Characterizing learning error

Learning errors can be classified into two categories. There are *tracking* errors, which arise because the target is not a solution of the true spiking network dynamics and *sampling* errors, which arise from encoding a continuous function with a finite number of spikes. We note that for a rate network, there would only be a tracking error. We quantified these learning errors as a function of the network and target time scales. The intrinsic time scale of spiking network dynamics was the synaptic decay constant $\tau_s$, and the time scale of target dynamics was the decay constant $\tau_c$ of OU noise. We used target patterns generated from OU noise since the trajectories have a predetermined time scale and their spatio-temporal patterns are sufficiently heterogeneous.

We systematically varied $\tau_s$ and $\tau_c$ from fast AMPA-like (~ 1 ms) to slow NMDA-like synaptic transmission (~ 100 ms) and trained the synaptic drive of networks with synaptic time scale $\tau_s$ to learn a set of OU trajectories with timescale $\tau_c$. The parameter scan revealed a learning regime, where the networks successfully encoded the target patterns, and two error-dominant regimes. The tracking error was prevalent when synapses were slow in comparison to target patterns, and the sampling error dominated when the synapses were fast (*Figure 6a*).

A network with a synaptic decay time $\tau_s = 200$ ms failed to follow rapid changes in the target patterns, but still captured the overall shape, when the target patterns had a faster time scale $\tau_c = 100$ ms (*Figure 6b*, Tracking error). This prototypical example showed that the synaptic dynamics were not fast enough to encode the target dynamics in the tracking error regime. With a faster synapse $\tau_s = 30$ ms, the synaptic drive was able to learn the identical target trajectories with high accuracy (*Figure 6b*, Learning). Note that although the target time scale ($\tau_c = 100$ ms) was significantly slower than the synaptic time scale ($\tau_s = 30$ ms), tuning the recurrent synaptic connections was sufficient for the network to generate slow network dynamics using fast synapses. This phenomenon was shown robustly in the learning regime in *Figure 4a* where learning occurred successfully for the parameters lying above the diagonal line ($\tau_c > \tau_s$). When the synapse was too fast $\tau_s = 5$ ms, however, the synaptic drive fluctuated around the target trajectories with high frequency (*Figure 6b*, Sampling error). This was a typical network response in the sampling error regime where discrete spikes with narrow width and large amplitude were summed to 'sample' the target synaptic activity.

To better understand how network parameters determined the learning errors, we mathematically analyzed the errors assuming that (1) target dynamics can be encoded if the quasi-static condition holds, and (2) the mean field description of the target dynamics is accurate (see Materials and methods, 'Analysis of learning error'). The learning errors were characterized as a deviation of these assumptions from the actual spiking network dynamics. We found that the tracking errors $\epsilon_{\text{track}}$ were substantial if the quasi-static condition was not valid, that is synapses were not fast enough for spiking networks to encode targets, and the sampling errors $\epsilon_{\text{sample}}$ occurred if the mean field description became inaccurate, that is discrete representation of targets in terms of spikes deviated from their continuous representation in terms of spiking rates. The errors were estimated to scale with

$$\epsilon_{\text{track}} \sim \tau_s/\tau_c, \epsilon_{\text{sample}} \sim 1/\sqrt{\tau_s N}, \tag{3}$$

which implied that tracking error can be controlled as long as synapses are relatively faster than target patterns, and the sampling error can be controlled by either increasing $\tau_s$ to stretch the width of individual spikes or increasing $N$ to encode the targets with more input spikes. The error estimates revealed the versatility of recurrent spiking networks to encode arbitrary patterns since $\epsilon_{\text{track}}$ can be reduced by tuning $\tau_s$ to be small enough and $\epsilon_{\text{sample}}$ can be reduced by increasing $N$ to be large enough. In particular, target signals substantially slower than the synaptic dynamics (i.e. $\tau_s/\tau_c \ll 1$) can be encoded reliably as long as the network size is large enough to represent the slow signals with filtered spikes that have narrow widths. Such slow dynamics were also investigated in randomly

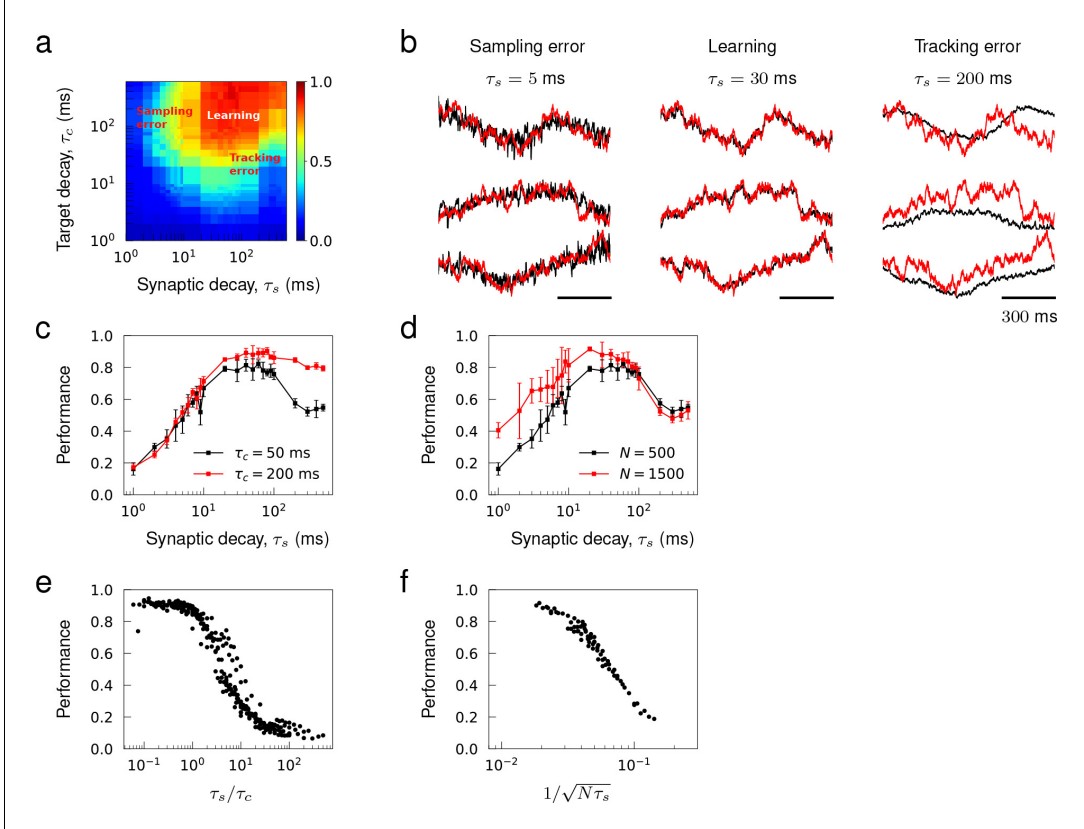

**Figure 6.** Sampling and tracking errors. Synaptic drive was trained to learn 1 s long trajectories generated from OU noise with decay time $\tau_c$. (a) Performance of networks of size $N = 500$ as a function of synaptic decay time $\tau_s$ and target decay time $\tau_c$. (b) Examples of trained networks whose responses show sampling error, tracking error, and successful learning. The target trajectories are identical and $\tau_c = 100$ ms. (c) Inverted 'U'-shaped curve as a function of synaptic decay time. Error bars show the s.d. of five trained networks of size $N = 500$. (d) Inverted 'U'-shaped curve for networks of sizes $N = 500$ and 1000 for $\tau_c = 100$ ms. (e) Network performance shown as a function of $\tau_s/\tau_c$ where the range of $\tau_s$ is from 30 ms to 500 ms and the range of $\tau_c$ is from 1ms to 500ms and $N = 1000$. (f) Network performance shown as a function of $1/\sqrt{N\tau_s}$ where the range of $\tau_s$ is from 1 ms to 30 ms, the range of $N$ is from 500 to 1000 and $\tau_c = 100$ ms.

DOI: https://doi.org/10.7554/eLife.37124.012

connected recurrent networks when coupling is strong (*Sompolinsky et al., 1988*; *Ostojic, 2014*) and reciprocal connections are over-represented (*Martí et al., 2018*).

We examined the performance of trained networks to verify if the theoretical results can explain the learning errors. The learning curve, as a function of $\tau_s$, had an inverted U-shape when both types of errors were present (*Figure 6c, d*). Successful learning occurred in an optimal range of $\tau_s$, and, consistent with the error analysis, the performance decreased monotonically with $\tau_s$ on the right branch due to increase in the tracking error while the performance increased monotonically with $\tau_s$ on the left branch due to decrease in the sampling error. The tracking error was reduced if target patterns were slowed down from $\tau_c = 50$ ms to $\tau_c = 200$ ms, hence decreased the ratio $\tau_s/\tau_c$. Then, the learning curve became sigmoidal, and the performance remained high even when $\tau_s$ was in the slow NMDA regime (*Figure 6c*). On the other hand, the sampling error was reduced if the network size was increased from $N = 500$ to 1500, which lifted the left branch of the learning curve (*Figure 6d*). Note that when two error regimes were well separated, changes in target time scale $\tau_c$ did not affect $\epsilon_{\text{sample}}$, and changes in network size $N$ did not affect $\epsilon_{\text{sample}}$, as predicted.

Finally, we condensed the training results over a wide range of target time scales in the tracking error regime (*Figure 6e*), and similarly condensed the training results over different network sizes in the sampling error regime (*Figure 6f*) to demonstrate that $\tau_s/\tau_c$ and $N\tau_s$ explained the overall performance in the tracking and sampling error regimes, respectively.

## Learning capacity increases with network size

It has been shown that a recurrent rate network's capability to encode target patterns deteriorates as a function of the length of time (*Laje and Buonomano, 2013*), but increase in network size can enhance its storage capacity (*Jaeger, 2001*; *White et al., 2004*; *Rajan et al., 2016*). Consistent with these results, we found that the performance of recurrent spiking networks to learn complex trajectories decreased with target length and improved with network size (*Figure 7a*).

To assess the storage capacity of spiking networks, we evaluated the maximal target length that can be encoded in a network as a function of network size. It was necessary to define the target length in terms of its 'effective length' to account for the fact that target patterns with the same length may have different effective length due to their temporal structures; for instance, OU noise with short temporal correlation times has more structure to be learned than a constant function. For target trajectories generated from an OU process with decay time $\tau_c$, we rescaled the target length $T$ with respect to $\tau_c$ and defined the effective length $\tilde{T} = T/\tau_c$. The capacity of a network was the maximal $\tilde{T}$ that can be successfully encoded in a network.

To estimate the maximal $\tilde{T}$, we trained networks of fixed size to learn OU trajectories while varying $T$ and $\tau_c$ (each panel in *Figure 7b*). Then, for each $\tau_c$, we found the maximal target length $T_{\max}$ that can be learned successfully, and estimated the maximal $\tilde{T}$ by finding a constant $\tilde{T}_{\max}$ that best fits the line $T_{\max} = \tilde{T}_{\max}\tau_c$ to training results (black lines in *Figure 7b*). *Figure 7c* shows that the learning capacity $\tilde{T}_{max}$ increases monotonically with the network size.

## Discussion

Our findings show that individual neurons embedded in a recurrent network can learn to produce complex activity by adjusting the recurrent synaptic connections. Most previous research on learning in recurrent neural networks focused on training the network outputs to perform useful computations and subsequently analyzed the recurrent activity in comparison with measured neuron activity (*Sussillo and Abbott, 2009*; *Sussillo et al., 2015*; *Sussillo and Barak, 2013*; *Wang et al., 2018*; *Chaisangmongkon et al., 2017*; *Remington et al., 2018*). In contrast to such output-centric approaches, our study takes a network-centric perspective and directly trains the activity of neurons within a network individually. Several studies have trained a rate-based network model to learn specific forms of target recurrent activity, such as innate chaotic dynamics (*Laje and Buonomano, 2013*), sequential activity (*Rajan et al., 2016*), and trajectories from a target network (*DePasquale et al., 2018*). In this study, we showed that the synaptic drive and spiking rate of a synaptically-coupled spiking network can be trained to follow arbitrary spatiotemporal patterns. The necessary ingredients for learning are that the spike train inputs to a neuron are weakly correlated

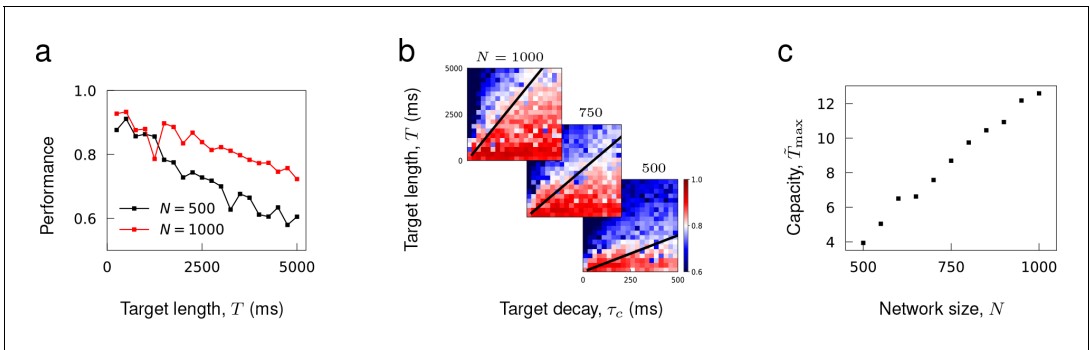

**Figure 7.** Capacity as a function of network size. (**a**) Performance of trained networks as a function of target length $T$ for networks of size $N = 500$ and 1000. Target patterns were generated from OU noise with decay time $\tau_c = 100$ ms. (**b**) Networks of fixed sizes trained on a range of target length and correlations. Color bar shows the Pearson correlation between target and actual synaptic drive. The black lines show the function $T_{max} = \tilde{T}_{max}\tau_c$ where $\tilde{T}_{max}$ was fitted to minimize the least square error between the linear function and maximal target length $T_{max}$ that can be successfully learned at each $\tau_c$. (**c**) Learning capacity $\tilde{T}_{max}$ shown as a function of network size.
DOI: https://doi.org/10.7554/eLife.37124.013

(i.e. heterogeneous target patterns), the synapses are fast enough (i.e. small tracking error), and the network is large enough (i.e. small sampling error and large capacity). We demonstrated that (1) a network consisting of excitatory and inhibitory neurons can learn to track its strongly fluctuating innate synaptic trajectories, and (2) are current spiking network can learn to reproduce the spiking rate patterns of an ensemble of cortical neurons involved in motor planning and movement.

Our scheme works because the network quickly enters a quasi-static state where the instantaneous firing rate of a neuron is a fixed function of the inputs (*Figure 3a, b*; *Equations 1, 2*). Learning fails if the synaptic time scale is slow compared to the time scale of the target, in which case the quasi-static condition is violated and the tracking error becomes large. There is a trade-off between tracking error and sampling noise; fast synapse can decrease the tracking error, but it also increases the sampling noise. Increasing the network size can decrease sampling noise without affecting the tracking error (*Figure 6e, f*; *Equation 3*). Therefore, analysis of learning error and simulations suggest that it is possible to learn arbitrarily complex recurrent dynamics by adjusting the synaptic time scale and network size.

An important structural property of our network model is that the synaptic inputs are summed linearly, which allows the synaptic activity to be trained using a recursive form of linear regression (*Sussillo and Abbott, 2009*; *Equation 6*). Linear summation of synaptic inputs is a standard assumption for many spiking network models (*van Vreeswijk et al., 1996*; *Renart et al., 2010*; *Brunel, 2000*; *Wang and Buzsáki, 1996*; *Rosenbaum et al., 2017*) and there is physiological evidence that linear summation is prevalent (*Cash and Yuste, 1998*; *Cash and Yuste, 1999*). Training the spiking rate, on the other hand, cannot take full advantage of the linear synapse due to the nonlinear current-to-transfer function (*Figure 1d, e*; *Equation 2*). The network is capable of following a wide repertoire of patterns because even though the network dynamics are highly nonlinear, the system effectively reduces to a linear system for learning. Moreover, learning capacity can be estimated using a simple solvability condition for a linear system. However, nonlinear dendritic processing has been widely observed (*Gasparini and Magee, 2006*; *Nevian et al., 2007*) and may have computational consequences (*Memmesheimer, 2010*; *Memmesheimer and Timme, 2012*; *Thalmeier et al., 2016*). It requires further investigation to find out whether a recurrent network with nonlinear synapses can be trained to learn arbitrary recurrent dynamics.

We note that our learning scheme does not train precise spike times; it either trains the spiking rate or the synaptic drive. The stimulus at the onset of the learning window attempts to set the network to a specific state, but due to the variability of the initial conditions the network states can only be set approximately close to each other across trials. Because of this discrepancy in network states at the onset, the spike times are not aligned precisely across trials. Hence, our learning scheme supports rate coding as opposed to spike coding. However, spike trains that have temporally irregular structure across neurons actually enhance the rate coding scheme by providing sufficient computational complexity to encode the target dynamics (Results, 'Sufficient conditions for learning'). In fact, all neurons in the network can be trained to follow the same target patterns as long as there is sufficient heterogeneity, for example noisy external input, and the neuron time constant is fast enough (*Figure 3—figure supplement 3*). We also note that the same learning scheme can also be used to train the recurrent dynamics of rate-based networks (*Figure 1—figure supplement 1*). In fact, the learning is more efficient in a rate network since there is no sampling error to avoid.

The RLS algorithm, as demonstrated in this and other studies (*Sussillo and Abbott, 2009*; *Sussillo et al., 2015*; *Laje and Buonomano, 2013*; *Rajan et al., 2016*; *DePasquale et al., 2018*; *Wang et al., 2018*), successfully generates desired outputs in a stable manner because the synaptic update rule contracts the network activity towards the target output, and the synaptic connections are adjusted while the network explores various states around the target trajectories. It would be interesting to examine more rigorously how such an iterative learning scheme turns a set of arbitrary functions into dynamic attractors to which the network dynamics converge transiently. Recent studies investigated how stable dynamics emerge when the read-outs of a rate-based network are trained to learn fixed points or continuous values (*Rivkind and Barak, 2017*; *Beer and Barak, 2018*). In addition, previous studies have investigated the mathematical relationship between the patterns of stored fixed points and the recurrent connectivity in simple network models (*Curto et al., 2013*; *Brunel, 2016*).

Although our results demonstrated that recurrent spiking networks have the capability to generate a wide range of repertoire of recurrent dynamics, it is unlikely that a biological network is using

this particular learning scheme. The learning rule derived from recursive least squares algorithm is very effective but is nonlocal in time, that is it uses the activity of all presynaptic neurons within the train time window to update synaptic weights. Moreover, each neuron in the network is assigned with a target signal and the synaptic connections are updated at a fast time scale as the error function is computed in a supervised manner. It would be of interest to find out whether more biologically plausible learning schemes, such as reward-based learning (*Fiete and Seung, 2006*; *Hoerzer et al., 2014*; *Miconi, 2017*) can lead to similar performance.

# Materials and methods

## Network of spiking neurons

We considered a network of $N$ randomly and sparsely connected quadratic integrate-and-fire neurons given by

$$\tau \dot{v}_i = I_i(t) + u_i(t) + v_i^2 \qquad (4)$$

where $v_i$ is a dimensionless variable representing membrane potential, $I_i(t)$ is an applied input, $u_i(t)$ is the total synaptic drive the neuron receives from other neurons in the recurrent network, and $\tau = 10$ ms is a neuron time constant. The threshold to spiking is zero input. For negative total input, the neuron is at rest and for positive input, $v_i$ will go to infinity or 'blow up' in finite time from any initial condition. The neuron is considered to spike at $v_i = \infty$ whereupon it is reset to $-\infty$ (*Ermentrout, 1996*; *Latham et al., 2000*).

To simulate the dynamics of quadratic integrate-and-fire neurons, we used its phase representation, that is theta neuron model, that can be derived by a simple change of variables, $v_i = \tan(\theta_i/2)$; its dynamics are governed by

$$\tau \dot{\theta}_i = 1 - cos\theta_i + (I_i(t) + u_i(t))(1 + cos\theta_i), \qquad (5)$$

where a spike is emitted when $\theta(t) = \pi$. The synaptic drive to a neuron obeys

$$\tau_s \dot{u}_i(t) = -u_i(t) + \sum_{j=1}^{N} W_{ij} s_j(t), \qquad (6)$$

where $s_j(t) = \sum_{t_j^k < t} \delta\left(t - t_j^k\right)$ is the spike train neuron $j$ generates up to time $t$, and $\tau_s$ is a synaptic time constant.

The recurrent connectivity $W_{ij}$ describes the synaptic coupling from neuron $j$ to neuron $i$. It can be any real matrix but in many of the simulations we use a random matrix with connection probability $p$, and the coupling strength of non-zero elements is modeled differently for different figures.

## Training recurrent dynamics

To train the synaptic and spiking rate dynamics of individual neurons, it is more convenient to divide the synaptic drive *Equation 6* into two parts; one that isolates the spike train of single neuron and computes its synaptic filtering

$$\tau_s \dot{r}_i(t) = -r_i(t) + s_i(t), \qquad (7)$$

and the other that combines all the presynaptic neurons' spiking activity and computes the synaptic drive

$$u_i(t) = \sum_{j=1}^{N} W_{ij} r_j(t). \qquad (8)$$

The synaptic drive $u_i$ and the filtered spike train $r_i$ are two measures of spiking activity that have been trained in this study. Note that *Equations 7 and 8* generate synaptic dynamics that are equivalent to *Equation 6*.

## Training procedure

We select $N$ target trajectories $f_1(t), ..., f_N(t)$ of length $T$ ms for a recurrent network consisting of $N$ neurons. We train either the synaptic drive or spiking rate of individual neuron $i$ to follow the target $f_i(t)$ over time interval $[0, T]$ for all $i = 1, ..., N$. External stimulus $I_i$ with amplitude sampled uniformly from $[-1, 1]$ is applied to neuron $i$ for all $i = 1, 2, ..., N$ for 100 ms immediately preceding the training to situate the network at a specific state. During training, the recurrent connectivity $W$ is updated every $\Delta t$ ms using a learning rule described below in order to steer the network dynamics toward the target dynamics. The training is repeated multiple times until changes in the recurrent connectivity stabilize.

## Training synaptic drive

Recent studies extended the RLS learning (also known as FORCE methods) developed in rate networks (*Sussillo and Abbott, 2009*) either directly (*Nicola and Clopath, 2017*) or indirectly using rate networks as an intermediate step (*DePasquale et al., 2016*; *Abbott et al., 2016*; *Thalmeier et al., 2016*) to train the output of spiking networks. Our learning rule uses the RLS learning but is different from previous studies in that (a) it trains the activity of individual neurons within a spiking network and (b) neurons are trained directly by adjusting the recurrent synaptic connections without using any intermediate networks. We modified the learning rule developed by Laje and Buonomano in a network of rate units (*Laje and Buonomano, 2013*) and also provided mathematical derivation of the learning rules for both the synaptic drive and spiking rates (see Materials and methods, 'Derivation of synaptic learning rules' for details).

When learning the synaptic drive patterns, the objective is to find recurrent connectivity $W$ that minimizes the cost function

$$C[W] = \int_0^T \frac{1}{2} ||\boldsymbol{f}(t) - \boldsymbol{u}(t)||_{L_2}^2 dt + \frac{\lambda}{2} ||W||_{L_2}^2, \tag{9}$$

which measures the mean-square error between the targets and the synaptic drive over the time interval $[0, T]$ plus a quadratic regularization term. To derive the learning rule, we use *Equation 8* to express $u$ as a function of $W$, view the synaptic connections $W_{i1}, ..., W_{iN}$ to neuron $i$ to be the read-out weights that determine the synaptic drive $u_i$, and apply the learning rule to the row vectors of $W$. To keep the recurrent connectivity sparse, learning occurs only on synaptic connections that are non-zero prior to training.

Let $\boldsymbol{w}_i(t)$ be the reduced row vector of $W(t)$ consisting of elements that have non-zero connections to neuron $i$ prior to training. Similarly, let $\boldsymbol{r}_i(t)$ be a (column) vector of filtered spikes of presynaptic neurons that have non-zero connections to neuron $i$. The synaptic update to neuron $i$ is

$$\boldsymbol{w}_i(t)^T = \boldsymbol{w}_i(t - \Delta t)^T + e_i(t) P(t) \boldsymbol{r}_i(t), \tag{10}$$

where the error term is

$$e_i(t) = f_i(t) - \boldsymbol{w}_i(t - \Delta t) \boldsymbol{r}_i(t) \tag{11}$$

and the inverse of the correlation matrix of filtered spike trains is

$$P(t) = P(t - \Delta t) - \frac{P(t - \Delta t) \boldsymbol{r}_i(t) \boldsymbol{r}_i(t)^T P(t - \Delta t)}{1 + \boldsymbol{r}_i(t)^T P(t - \Delta t) \boldsymbol{r}_i(t)}, P(0) = \lambda^{-1} I. \tag{12}$$

Finally, $W(t)$ is obtained by concatenating the row vectors $\boldsymbol{w}_i(t), i = 1, ..., N$.

## Training spiking rate

To train the spiking rate of neurons, we approximate the spike train $s_i(t)$ of neuron $i$ with its spiking rate $\phi(u_i(t) + I_i)$ where $\phi$ is the current-to-rate transfer function of theta neuron model. For constant input,

$$\phi_1(x) = \pi^{-1}\sqrt{[x]_+} \text{ where } [x]_+ = \max(x,0), \tag{13}$$

and for noisy input

$$\phi_2(x) = \frac{1}{\pi}\sqrt{c\log(1+e^{x/c})}. \tag{14}$$

Since $\phi_2$ is a good approximation of $\phi_1$ and has a smooth transition around $x = 0$, we used $\phi \equiv \phi_2$ with $c = 0.1$ (**Brunel and Latham, 2003**). The objective is to find recurrent connectivity $W$ that minimizes the cost function

$$C[W] = \int_0^T \frac{1}{2}||\boldsymbol{f}(t) - \phi(W\boldsymbol{r}(t) + \boldsymbol{I})||_{L_2}^2 dt + \frac{\lambda}{2}||W||_{L_2}^2. \tag{15}$$

If we define $w_i$ and $r_i$ as before, we can derive the following synaptic update to neuron $i$

$$\boldsymbol{w}_i^T(t) = \boldsymbol{w}_i^T(t-\Delta t) + e_i(t)P(t)\tilde{\boldsymbol{r}}_i(t), \tag{16}$$

where the error term is

$$e_i(t) = f_i(t) - \phi(\boldsymbol{w}_i(t-\Delta t)\boldsymbol{r}_i(t) + I_i) \tag{17}$$

and

$$P(t) = P(t-\Delta t) - \frac{P(t-\Delta t)\tilde{\boldsymbol{r}}_i(t)\tilde{\boldsymbol{r}}_i(t)^T P(t-\Delta t)}{1 + \tilde{\boldsymbol{r}}_i(t)^T P(t-\Delta t)\tilde{\boldsymbol{r}}_i(t)}, P^0 = \lambda^{-1}I. \tag{18}$$

(see Materials and methods, 'Derivation of synaptic learning rules' for details). Note that the nonlinear effects of the transfer function is included in

$$\tilde{\boldsymbol{r}}_i(t) = \phi'(u_i(t) + I_i)\boldsymbol{r}_i(t), \tag{19}$$

which scales the spiking activity of neuron $i$ by its gain function $\phi'$.

As before, $W(t)$ is obtained by concatenating the row vectors $\boldsymbol{r}_i(t), i = 1,...,N$.

## Simulation parameters
### *Figure 1*
A network of $N = 200$ neurons was connected randomly with probability $p = 0.3$ and the coupling strength was drawn from a Normal distribution with mean 0 and standard deviation $\sigma/\sqrt{Np}$ with $\sigma = 4$. In addition, the average of all non-zero synaptic connections to a neuron was subtracted from the connections to the neuron such that the summed coupling strength was precisely zero. Networks with balanced excitatory and inhibitory connections produced highly fluctuating synaptic and spiking activity in all neurons. The synaptic decay time was $\tau_s = 20$ ms.

The target functions for the synaptic drive (**Figure 1b**) were sine waves $f(t) = A\sin(2\pi(t - T_0)/T_1)$ where the amplitude $A$, initial phase $T_0$, and period $T_1$ were sampled uniformly from $[0.5, 1.5]$, $[0, 1000\text{ms}]$ and $[300\text{ms}, 1000\text{ms}]$, respectively. We generated $N$ distinct target functions of length $T = 1000$ ms. The target functions for the spiking rate (**Figure 1d**) were $\pi^{-1}\sqrt{[f(t)]_+}$ where $f(t)$ were the same synaptic drive patterns that have been generated.

Immediately before each training loop, every neuron was stimulated for 50 ms with constant external stimulus that had random amplitude sampled from $[-1, 1]$. The same external stimulus was used across training loops. The recurrent connectivity was updated every $\Delta t = 2$ ms during training using the learning rule derived from RLS algorithm and the learning rate was $\lambda = 1$. After training, the network was stimulated with the external stimulus to evoke the trained patterns. The performance was measured by calculating the average Pearson correlation between target functions and the evoked network response.

## Figure 2

The initial network and target functions were generated as in *Figure 1* using the same parameters, but now the target functions consisted of two sets of $N$ sine waves. To learn two sets of target patterns, the training loops alternated between two patterns, and immediately before each training loop, every neuron was stimulated for 50 ms with constant external stimuli that had random amplitudes, using a different stimulus for each pattern. Each target pattern was trained for 100 loops (i.e. total 200 training loops), synaptic update was every $\Delta t = 2$ ms, and the learning rate was $\lambda = 10$. To evoke one of the target patterns after training, the network was stimulated with the external stimulus that was used to train that target pattern.

The network consisted of $N = 500$ neurons. The initial connectivity was sparsely connected with connection probability $p = 0.3$ and coupling strength was sampled from a Normal distribution with mean 0 and standard deviation $\sigma/\sqrt{Np}$ with $\sigma = 1$. The synaptic decay time was $\tau_s = 20$ ms.

We considered three families of target functions with length $T = 1000$ ms. The complex periodic functions were defined as a product of two sine waves $f(t) = A \sin(2\pi(t - T_0)/T_1) \sin(2\pi(t - T_0)/T_2)$ where $A$, $T_0$, $T_1$ and $T_2$ were sampled randomly from intervals $[0.5, 1.5]$, $[0, 1000\,\text{ms}]$, $[500\,\text{ms}, 1000\,\text{ms}]$, and $[100\,\text{ms}, 500\,\text{ms}]$, respectively. The chaotic rate activity was generated from a network of $N$ randomly connected rate units, $\tau \dot{x}_i = -x_i + \sum_{j=1}^{N} M_{ij} h(x_j)$ where $\tau = 40$ ms, $h(x) = \pi^{-1}\sqrt{[x]_+}$ and $M_{ij}$ is non-zero with probability $p = 0.3$ and is drawn from Gaussian distribution with mean zero and standard deviation $g/\sqrt{Np}$ with $g = 5$. The Ornstein-Ulenbeck process was obtained by simulating, $\tau_c \dot{x} = -x + s\xi(t)$, $N$ times with random initial conditions and different realizations of the white noise $\xi(t)$ satisfying $\langle \xi \rangle = 0$ and $\langle \xi(t)\xi(t') \rangle = \delta(t - t')$. The decay time constant was $\tau_c = 200$ ms, and the amplitude of target function was determined by $s = 0.3$.

The recurrent connectivity was updated every $\Delta t = 2$ ms during training, the learning rate was $\lambda = 1$, and the training loop was repeated 30 times.

## Figure 4

A balanced network had two populations where the excitatory population consisted of $(1 - f)N$ neurons and the inhibitory population consisted of $fN$ neurons with ratio $f = 0.2$ and network size $N = 1000$. Each neuron received $p(1 - f)N$ excitatory connections with strength $J$ and $pfN$ inhibitory connections with strength $-gJ$ from randomly selected excitatory and inhibitory neurons. The connection probability was set to $p = 0.1$ to have sparse connectivity. The relative strength of inhibition to excitation $g$ was set to 5 so that the network was inhibition dominant (*Brunel, 2000*). In *Figure 4a–h*, the initial coupling strength $J = 6$ and synaptic decay time $\tau_s = 60$ ms were adjusted to be large enough, so that the synaptic drive and spiking rate of individual neurons fluctuated strongly and slowly prior to training.

After running the initial network that started at random initial conditions for 3 s, we recorded the synaptic drive of all neurons for 2 s to harvest target trajectories that are innate to the balanced network. Then, the synaptic drive was trained to learn the innate trajectories, where synaptic update occurred every 10 ms, learning rate was $\lambda = 10$ and training loop was repeated 40 times. To respect Dale's Law while training the network, we did not modify the synaptic connections if the synaptic update reversed the sign of original connections, either from excitatory to inhibitory or from inhibitory to excitatory. Moreover, the synaptic connections that attempted to change their signs were excluded in subsequent trainings. In bf *Figure 4h*, the initial and trained connectivity matrices were normalized by a factor $\sqrt{\left[(1 - f)J^2 + f(gJ)^2\right](1 - p)}$ so that the spectral radius of the initial connectivity matrix is approximately 1, then we plotted the eigenvalue spectrum of the normalized matrices.

In *Figure 4i*, the coupling strength $J$ was scanned from 1 to 6 in increments of 0.25, and the synaptic decay time $\tau_s$ was scanned from 5 ms to 100 ms in increments of 5 ms. To measure the accuracy of quasi-static approximation in untrained networks, we simulated the network dynamics for each pair of $J$ and $\tau_s$, then calculated the average Person correlation between the predicted synaptic drive (*Equation 1*) and the actual synaptic drive. To measure the performance of trained networks, we repeated the training 10 times using different initial network configurations and innate trajectories, and calculated the Pearson correlation between the innate trajectories and the evoked network response for all 10 trainings. The heat map shows the best performance out of 10 trainings for each pair, $J$ and $\tau_s$.

## Figure 5

The initial connectivity was sparsely connected with connection probability $p = 0.3$ and the coupling strength was sampled from a Normal distribution with mean 0 and standard deviation $\sigma/\sqrt{Np}$ with $\sigma = 1$. The synaptic decay time was $\tau_s = 50$ ms. There were in total $N$ neurons in the network model, of which $N_{cor}$ neurons, called cortical neurons, were trained to learn the spiking rate patterns of cortical neurons, and $N_{aux}$ neurons, called auxiliary neurons, were trained to learn trajectories generated from OU process.

We used the trial-averaged spiking rates of neurons recorded in the anterior lateral motor cortex of mice engaged in motor planning and movement that lasted 4600 ms (*Li et al., 2015*). The data was available from the website CRCNS.ORG (*Li et al., 2014*). We selected $N_{cor} = 227$ neurons from the data set, whose average spiking rate during the behavioral task was greater than 5 Hz. Each cortical neuron in the network model was trained to learn the spiking rate pattern of one of the real cortical neurons.

To generate target rate functions for the auxiliary neurons, we simulated an OU process, $\tau_c \dot{x}(t) = -x(t) + s\xi(t)$, with $\tau_c = 800$ ms and $s = 0.1$, then converted into spiking rate $\phi([x(t)]_+)$ and low-pass filtered with decay time $\tau_s$ to make it smooth. Each auxiliary neuron was trained on 4600 ms-long target rate function that was generated with a random initial condition.

## Figure 6 and 7

Networks consisting of $N = 500$ neurons with no initial connections and synaptic decay time $\tau_s$ were trained to learn OU process with decay time $\tau_c$ and length $T$. In *Figure 6*, target length was fixed to $T = 1000$ ms while the time constants $\tau_s$ and $\tau_c$ were varied systematically from $10^0$ ms to $5 \cdot 10^2$ ms in log-scale. The trainings were repeated five times for each pair of $\tau_s$ and $\tau_c$ to find the average performance. In *Figure 7*, the synaptic decay time was fixed to $\tau_s = 20$ ms and $T$ was scanned from 250 ms to 5000 ms in increments of 250 ms, $\tau_c$ was scanned from 25 ms to 500 ms in increments of 25 ms, and $N$ was scanned from 500 to 1000 in increments of 50.

To ensure that the network connectivity after training is sparse, synaptic learning occurred only on connections that were randomly selected with probability $p = 0.3$ prior to training. Recurrent connectivity was updated every $\Delta t = 2$ ms during training, learning rate was $\lambda = 1$, and training loop was repeated 30 times. The average Pearson correlation between the target functions and the evoked synaptic activity was calculated to measure the network performance after training.

## Derivation of synaptic learning rules

Here, we derive the synaptic update rules for the synaptic drive and spiking rate trainings, *Equations 10 and 16*. We use RLS algorithm (*Haykin, 1996*) to learn target functions $f_i(t), i = 1, 2, ..., N$ defined on a time interval $[0, T]$, and the synaptic update occurs at evenly spaced time points, $0 = t^0 \leq t^1 ... \leq t^K = T$.

In the following derivation, super-script $k$ on a variable $X_i^k$ implies that $X$ is evaluated at $t^k$, and the sub-script $i$ implies that $X$ pertains to neuron $i$.

### Training synaptic drive

The cost function measures the discrepancy between the target functions $f_i(t)$ and the synaptic drive $u_i(t)$ for all $i = 1, ..., N$ at discrete time points $t_0, ..., t_K$,

$$C[W] = \frac{1}{2} \sum_{k=0}^{K} \|\boldsymbol{f}^k - \boldsymbol{u}^k\|_{L_2}^2 + \frac{\lambda}{2} \|W\|_{L_2}^2. \tag{20}$$

The Recursive Least Squares (RLS) algorithm solves the problem iteratively by finding a solution $W^n$ to *Equation 20* at $t^n$ and updating the solution at next time step $t^{n+1}$. We do not directly find the entire matrix $W^n$, but find each row of $W^n$, that is synaptic connections to each neuron $i$ that minimize the discrepancy between $u_i$ and $f_i$, then simply combine them to obtain $W^n$.

To find the $i^{th}$ row of $W^n$, we denote it by $w_i^n$ and rewrite the cost function for neuron $i$ that evaluates the discrepancy between $f_i(t)$ and $u_i(t)$ on a time interval $[0, t^n]$,

$$C[w_i^n] = \frac{1}{2}\sum_{k=0}^{n}\left(f_i^k - w_i^n \cdot r^k\right)^2 + \frac{\lambda}{2}\|w_i^n\|_{L_2}^2. \tag{21}$$

Calculating the gradient and setting it to 0, we obtain

$$0 = \nabla_{w_i^n} C = -\sum_{k=1}^{n}\left(\hat{u}_i^k - w_i^n \cdot r^k\right)r^k + \lambda w_i^n$$

We express the equation concisely as follows.

$$[R^n + \lambda I]w_i^n = q^n$$

$$R^n = \sum_{k=1}^{n} r^k (r^k)^T, q^n = \sum_{k=1}^{n} \hat{u}_i^k r^k. \tag{22}$$

To find $w_i^n$ iteratively, we rewrite *Equation 22* up to $t^{n-1}$,

$$[R^{n-1} + \lambda I]w_i^{n-1} = q^{n-1}, \tag{23}$$

and subtract *Equations 23 and 24* to obtain

$$[R^n + \lambda I]\left[w_i^n - w_i^{n-1}\right] + r^n(r^n)^T w_i^{n-1} = \hat{u}_i^n r^n. \tag{24}$$

The update rule for $w_i^n$ is then given by

$$w_i^n = w_i^{n-1} + e_i^n [R^n + \lambda I]^{-1} r^n, \tag{25}$$

where the error term is

$$e_i^n = f_i^n - r^n \cdot w_i^{n-1}. \tag{26}$$

The matrix inverse $P^n = [R^n + \lambda I]^{-1}$ can be computed iteratively

$$P^n = P^{n-1} - \frac{P^{n-1} r^n (r^n)^T P^{n-1}}{1 + (r^n)^T P^{n-1} r^n}, P^0 = \lambda^{-1} I,$$

using the matrix identity

$$\left(A + rr^T\right)^{-1} = A^{-1} - \frac{A^{-1} rr^T A^{-1}}{1 + r^T A^{-1} r}.$$

## Training spiking rate

To train the spiking rate of neurons, we approximate the spike train $s_i(t)$ of neuron $i$ with its spiking rate $\phi(u_i(t) + I_i)$ where $\phi$ is the current-to-rate transfer function of theta neuron model. For constant input,

$$\phi_1(x) = \pi^{-1}\sqrt{[x]_+} \text{ where } [x]_+ = max(x, 0), \tag{27}$$

and for noisy input

$$\phi_2(x) = \frac{1}{\pi}\sqrt{c\log(1 + e^{x/c})}. \tag{28}$$

Since $\phi_2$ is a good approximation of $\phi_1$ and has a smooth transition around $x = 0$, we used $\phi \equiv \phi_2$ with $c = 0.1$ (*Brunel and Latham, 2003*).

If the synaptic update occurs at discrete time points, $t^0, ..., t^K$, the objective is to find recurrent connectivity $W$ that minimizes the cost function

$$C[W] = \frac{1}{2}\sum_{k=0}^{K}\left\|f^k(t) - \phi\left(Wr^k(t) + I\right)\right\|_{L_2}^2 + \frac{\lambda}{2}\|W\|_{L_2}^2. \tag{29}$$

As in training the synaptic drive, we optimize the following cost function to train each row of $W^n$ that evaluates the discrepancy between the spiking rate of neuron $i$ and the target spiking rate $f_i$ over a time interval $[0, t^n]$,

$$C\left[w_i^n\right] = \frac{1}{2} \sum_{k=1}^{n} \left(f_i^k - \phi\left(\boldsymbol{w}_i^n \cdot \boldsymbol{r}^k + I_i^k\right)\right)^2 + \frac{\lambda}{2} |\boldsymbol{w}_i^n|^2. \tag{30}$$

Calculating the gradient and setting it to zero, we obtain

$$0 = \nabla_{w_i^n} C = -\sum_{k=1}^{n} \left[f_i^k - \phi\left(\boldsymbol{w}_i^n \cdot \boldsymbol{r}^k + I_i^k\right)\right] \tilde{\boldsymbol{r}}_i^k + \lambda w_i^n. \tag{31}$$

where

$$\tilde{\boldsymbol{r}}_i^k = \phi'\left(u_i^k + I_i^k\right) \boldsymbol{r}^k \tag{32}$$

is the vector of filtered spike trains scaled by the gain of neuron $i$. Note that when evaluating $\phi'$ in *Equation 32*, we use the approximation $u_i^k \approx \boldsymbol{w}_i^n \cdot \boldsymbol{r}^k$ to avoid introducing nonlinear functions of $\boldsymbol{w}_i^n$.

To find an update rule for $w_i^n$, we rewrite *Equation 31* up to $t_{n-1}$,

$$0 = -\sum_{k=1}^{n-1} \left[f_i^k - \phi\left(\boldsymbol{w}_i^{n-1} \cdot \boldsymbol{r}^k + I_i^k\right)\right] \tilde{\boldsymbol{r}}_i^k + \lambda \boldsymbol{w}_i^{n-1}, \tag{33}$$

and subtract *Equations 31 and 33* and obtain

$$0 = \sum_{k=1}^{n} \left[\phi\left(\boldsymbol{w}_i^n \cdot \boldsymbol{r}^k + I_i^k\right) - \phi\left(\boldsymbol{w}_i^{n-1} \cdot \boldsymbol{r}^k + I_i^k\right)\right] \tilde{\boldsymbol{r}}_i^k \tag{34}$$

$$- \left[f_i^n - \phi\left(\boldsymbol{w}_i^{n-1} \cdot \boldsymbol{r}^n + I_i^n\right)\right] \tilde{\boldsymbol{r}}_i^k + \lambda \left[\boldsymbol{w}_i^n - \boldsymbol{w}_i^{n-1}\right].$$

Since $w_i^{n-1}$ is updated by small increment, we can approximate the first line in *Equation 34*,

$$\phi\left(\boldsymbol{w}_i^n \cdot \boldsymbol{r}^k + I_i^k\right) - \phi\left(\boldsymbol{w}_i^{n-1} \cdot \boldsymbol{r}^k + I_i^k\right) \approx \left[\boldsymbol{w}_i^n - \boldsymbol{w}_i^{n-1}\right] \cdot \tilde{\boldsymbol{r}}_i^k \tag{35}$$

where we use the approximation $u_i^k \approx \boldsymbol{w}_i^n \cdot r^k$ as before to evaluate the derivative $\phi'$. Substituting *Equation 35* to *Equation 34*, we obtain the update rule

$$\boldsymbol{w}_i^n = \boldsymbol{w}_i^{n-1} + e_i^n \left[R^n + \lambda I\right]^{-1} \tilde{\boldsymbol{r}}_i^n, \tag{36}$$

where the error is

$$e_i^n = f_i^n - \phi\left(\boldsymbol{w}_i^{n-1} \cdot \boldsymbol{r}^n + I_i^n\right), \tag{37}$$

and the correlation matrix of the normalized spiking activity is

$$R^n = \sum_{k=1}^{n} \tilde{\boldsymbol{r}}_i^k \left(\tilde{\boldsymbol{r}}_i^k\right)^T. \tag{38}$$

As shown above, the matrix inverse $P^n = \left[R^n + \lambda I\right]^{-1}$ can be computed iteratively,

$$P^n = P^{n-1} - \frac{P^{n-1} \tilde{\boldsymbol{r}}_i^n \left(\tilde{\boldsymbol{r}}_i^n\right)^T P^{n-1}}{1 + \left(\tilde{\boldsymbol{r}}_i^n\right)^T P^{n-1} \tilde{\boldsymbol{r}}_i^n}, P^0 = \lambda^{-1} I.$$

## Mean field description of the quasi-static dynamics

We say that a network is in a quasi-static state if the synaptic drive to a neuron changes sufficiently slower than the dynamical time scale of neurons and synapses. Here, we use a formalism developed by *Buice and Chow (2013)* and derive *Equations 1 and 2*, which provide a mean field description of the synaptic and spiking rate dynamics of neurons in the quasi-static state.

First, we recast single neuron dynamic *Equation 5* in terms of the empirical distribution of neuron's phase $\eta_i(\theta, t) = \delta(\theta_i(t) - \theta)$. Since the number of neurons in the network is conserved, we can write the Klimontovich equation for the phase distribution

$$\partial_t \eta_i(\theta, t) + \partial_\theta [F(\theta, u_i + I_i)\eta_i(\theta, t)] = 0 \tag{39}$$

where $F(\theta, I) = 1 - cos\theta + I(1 + cos\theta)$. The synaptic drive *Equation 6* can be written in the form

$$\tau_s \dot{u}_i(t) = -u_i(t) + 2\sum_{j=1}^{N} W_{ij}\eta_j(\pi, t) \tag{40}$$

since $s_j(t) = \eta_j(\pi, t)\dot{\theta}|_{\theta_j = \pi}$ and $\dot{\theta}_j|_{\theta_j = \pi} = 2$ for a theta neuron model. *Equation 39*, together with *Equation 40*, fully describes the network dynamics.

Next, to obtain a mean field description of the spiking dynamics, we take the ensemble average prepared with different initial conditions and ignore the contribution of higher order moments resulting from nonlinear terms $\langle u_i \eta_i \rangle$. Then we obtain the mean field equation

$$\partial_t \rho(\theta, t) + \partial_\theta [F(\theta, U_i + I_i)\rho_i(\theta, t)] = 0 \tag{41}$$

$$\tau_s \dot{U}_i = -U_i + 2\sum_{j=1}^{N} W_{ij}\rho_j(\pi, t). \tag{42}$$

where $\langle u_i \rangle = U_i$ and $\langle \eta_i \rangle = \rho_i$. We note that the mean field *Equations 41 and 42* provide a good description of the trained network dynamics because $W$ learns over repeatedly training trials and starting at random initial conditions, to minimize the error between target trajectories and actual neuron activity.

Now, we assume that the temporal dynamics of synaptic drive and neuron phase can be suppressed in the quasi-static state,

$$\tau_s \dot{U}_i \approx 0, \partial_t \rho \approx 0. \tag{43}$$

Substituting *Equation 43* to *Equation 41*, but allowing $U(t)$ to be time-dependent, we obtain the quasi-static solution of phase density

$$\rho_i(\theta, t) = \frac{\sqrt{[U_i(t) + I_i]_+}}{\pi[1 - cos\theta + (U_i(t) + I_i)(1 + cos\theta)]}, \tag{44}$$

$$\int_{-\pi}^{\pi} \rho_i(\theta)d\theta = 1$$

$$\phi(U_i(t) + I_i) = 2\rho_i(\pi, t) = \sqrt{[U_i(t) + I_i]_+}/\pi, \tag{45}$$

the current-to-rate transfer function of a theta neuron model. Substituting *Equation 43* and *Equation 45* to *Equation 42*, we obtain a quasi-static solution of the synaptic drive

$$U_i(t) = \sum_{j=1}^{N} W_{ij}\phi(U_j(t) + I_j). \tag{46}$$

If we define the spiking rate of a neuron as $R_i(t) = \phi(U_i + I_i)$, we immediately obtain

$$R_i(t) = \phi\left(\sum_{j=1}^{N} W_{ij}R_j + I_i\right). \tag{47}$$

## Analysis of learning error

In this section, we identify and analyze two types of learning errors, assuming that for sufficiently heterogeneous targets, (1) the learning rule finds a recurrent connectivity $W$ that can generate target

patterns if the quasi-static condition holds, and (2) the mean field description of the spiking network dynamics is accurate due to the error function and repeated training trials. These assumptions imply that *Equations 46 and 47* hold for the target patterns $U_i(t)$ and the trained $W$. We show that learning errors arise when our assumptions become inaccurate, hence the network dynamics described by *Equations 46 and 47* deviate from the actual spiking network dynamics. As we will see, tracking error is prevalent if the target is not an exact solution of the mean field dynamics (i.e. quasi-static approximation fails), and the sampling error dominates if the discrete spikes do not accurately represent continuous targets (i.e. mean field approximation fails).

Suppose we are trying to learn a target $\hat{u}_i$ which obeys an Ornstein-Ulenbeck process

$$\left(\tau_c \frac{d}{dt} + 1\right)\hat{u}_i = \xi_i(t) \tag{48}$$

on a time interval $0 < t < T$ where $\xi_i(t)$ are independent white noise with zero mean and variance $\sigma^2$. The time constant $\tau_c$ determines the temporal correlation of a target trajectory. In order for perfect training, the target dynamics (*Equation 48*) needs to be compatible with the network dynamics (*Equation 6*); in other words, there must exist a recurrent connectivity $W$ such that the following equation

$$\left(\tau_s \frac{d}{dt} + 1\right)\hat{u}_i(t) = \sum_{j=1}^{N} W_{ij} s\left[\hat{u}_j(t)\right] \tag{49}$$

obtained by substituting the solution of *Equation 48* into *Equation 6* must hold for $0 < t < T$. Here, $s\left[\hat{u}_j(t)\right]$ maps the synaptic drive $\hat{u}_j(t)$ to the entire spike train $s_j(t)$.

It is very difficult to find $W$ that may solve *Equation 49* exactly since it requires fully understanding the solution space of a high-dimensional system of nonlinear ordinary differential equations. Instead, we assume that the target patterns are quasi-static and the learning rule finds a recurrent connectivity $W$ that satisfies

$$\hat{u}_i(t) = \sum_{j=1}^{N} W_{ij} \phi\left(\hat{u}_j(t)\right). \tag{50}$$

We then substitute *Equation 50* to *Equation 49* to estimate how the quasi-static mean field dynamics deviate from the actual spiking network dynamics. A straightforward calculation shows that

$$\hat{u}_i(t) - \sum_{j=1}^{N} W_{ij} \phi\left(\hat{u}_j(t)\right) + \epsilon_{\text{track}} + \epsilon_{\text{sample}} = 0 \tag{51}$$

where we define the tracking and sampling errors as

$$\epsilon_{\text{track}} = \tau_s \frac{d\hat{u}_i}{dt} \tag{52}$$

and

$$\epsilon_{\text{sample}} = \sum_{j=1}^{N} W_{ij}\left(\phi\left(\hat{u}_j(t)\right) - s\left[\hat{u}_j(t)\right]\right) \tag{53}$$

on the time interval $0 < t < T$.

## Tracking error

From its definition, $\epsilon_{\text{track}}$ captures the deviation of the quasi-static solution (*Equation 50*) from the exact solution of the mean field description obtained when $\epsilon_{\text{sample}} = 0$. $\epsilon_{\text{track}}$ becomes large if the quasi-static condition (*Equation 43*) fails and, in such network state, the synaptic dynamic is not able to 'track' the target patterns, thus learning is obstructed. In the following, we estimate $\epsilon_{\text{track}}$ in terms of two time scales $\tau_s$ and $\tau_c$.

First, we take the Fourier transform of *Equation 52* and obtain

$$F[\epsilon_{\text{track}}](\omega) = i\tau_s\omega \cdot F[\hat{u}](\omega). \tag{54}$$

Next, normalize $F[\epsilon_{\text{track}}]$ with respect to $F[\hat{u}]$ to estimate the tracking error for target patterns with different amplitudes, then compute the power of normalized tracking error.

$$\frac{1}{\Omega}\int_0^\Omega \left\| \frac{F[\epsilon_{\text{track}}]}{F[\hat{u}]} \right\| d\omega = \frac{1}{2}\tau_s\Omega|_{\Omega=\Omega_c} = \frac{1}{4\pi}\frac{\tau_s}{\tau_c} \tag{55}$$

where $\Omega_c = 1/(2\pi\tau_c)$ is the cut-off frequency of the power spectrum of a Gaussian process, $S_{GP}(\omega) = \sigma^2\tau_c^2/\left(1 + 4\pi^2\tau_c^2\omega\right)$. Thus, the tracking error scales with $\tau_s/\tau_c$.

## Sampling error

$\epsilon_{\text{sample}}$ captures how the actual representation of target patterns in terms of spikes deviates from their continuous representation in terms of rate functions. In the following, we estimate $\epsilon_{extsample}$ in terms of $\tau_s$ and $N$ under the assumption that the continuous representation provides an accurate description of the target patterns.

We low-pass filtered $\epsilon_{\text{sample}}$ to estimate the sampling error since the synaptic drive (i.e. the target variable in this estimate) is a $W$ weighted sum of filtered spikes with width that scales with $\tau_s$. If the spike trains of neurons are uncorrelated (i.e. cross product terms are negligible),

$$\text{Var}\left[\epsilon_{\text{sample}}^{\text{filtered}}\right] = \sum_{j=1}^N W_{ij}^2 \left\langle \left(\bar{r}_j - r_j(t)\right)^2 \right\rangle \tag{56}$$

where $r_j(t)$ is the filtered spike train and $\bar{r}_j = \langle r_j(t) \rangle = \frac{1}{\Delta t}\int_{t_k}^{t_{k+1}} r_j(s)ds$ is the empirical estimate of mean spiking rate on a short time interval.

First, we calculate the fluctuation of filtered spike trains under the assumption that a neuron generates spikes sparsely, hence the filtered spikes are non-overlapping. Let $s_j(t) = \sum_k \delta\left(t - t_j^k\right)$ be a spike train of neuron $j$ and the filtered spike train $r_j(t) = \frac{1}{\tau_s}\sum_k \exp\left(-\left(t - t_j^k\right)/\tau_s\right)H\left(t - t_j^k\right)$. Then, the rate fluctuation of neuron $j$ is

$$\left\langle \left(r_j(t) - \bar{r}_j\right)^2 \right\rangle = \left\langle r_j^2(t) \right\rangle - \bar{r}^2 \tag{57}$$

$$= \frac{1}{\tau_s^2}\sum_k \left\langle exp\left(-2\left(t - t_j^k\right)/\tau_s\right)H\left(t - t_j^k\right)\right\rangle - \bar{r}^2 \tag{58}$$

$$= \bar{r}_j\left(\frac{1}{2\tau_s} - \bar{r}_j\right) \tag{59}$$

where $k$ is summed over the average number of spikes, $\bar{r}_j\Delta t$, generated in the time interval of length $\Delta t$.

Next, to estimate the effect of network size on the sampling error, we examined *Equation 50* and observed that $O(W) \sim 1/N$. This follows from that, for pre-determined target patterns, $O(U), O(\phi(U)) \sim 1$ regardless of the network size, hence $O(W)$ must scale with $1/N$ in order for both sides of the equation to be compatible. If the network is dense, that is the number of synaptic connections to a neuron is $pN$ on average, then the sampling error scales as follows.

$$O\left(\text{Var}\left[\epsilon_{\text{sample}}^{\text{filtered}}\right]\right) \sim \sum_{j=1}^N O\left(W_{ij}^2\right)O\left(\left\langle \left(\bar{r}_j - r_j(t)\right)^2 \right\rangle\right) \sim \frac{1}{\tau_s N} \tag{60}$$

## Acknowledgements

This research was supported [in part] by the Intramural Research Program of the NIH, The National Institute of Diabetes and Digestive and Kidney Diseases (NIDDK).

## Additional information

### Funding

| Funder | Grant reference number | Author |
|---|---|---|
| National Institute of Diabetes and Digestive and Kidney Diseases | Intramural Research Program | Christopher M Kim<br>Carson C Chow |

The funders had no role in study design, data collection and interpretation, or the decision to submit the work for publication.

### Author contributions

Christopher M Kim, Data curation, Software, Formal analysis, Investigation, Visualization, Methodology, Writing—original draft, Writing—review and editing; Carson C Chow, Conceptualization, Supervision, Funding acquisition, Investigation, Methodology, Writing—review and editing

### Author ORCIDs

Christopher M Kim (ID) http://orcid.org/0000-0002-1322-6207
Carson C Chow (ID) http://orcid.org/0000-0003-1463-9553

### Decision letter and Author response

Decision letter https://doi.org/10.7554/eLife.37124.016
Author response https://doi.org/10.7554/eLife.37124.017

## Additional files

### Supplementary files

• Transparent reporting form
DOI: https://doi.org/10.7554/eLife.37124.014

Example computer code that trains recurrent spiking networks is available at https://github.com/chrismkkim/SpikeLearning (copy archived at https://github.com/elifesciences-publications/SpikeLearning)

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
