## [Decision Letter]

Thank you for submitting your article "Learning recurrent dynamics in spiking networks" for consideration by *eLife*. Your article has been reviewed by Timothy Behrens as the Senior Editor and two reviewers, one of whom is the Reviewing Editor. One of the reviewers, Brian DePasquale, has agreed to reveal his identity.

The Reviewing Editor has drafted this decision to help you prepare a revised submission.

Summary:

The focus of most (maybe all?) research on RNNs has been on output: how do you train recurrent networks to produce a low dimensional (compared to the dimensionality of the network) target function? This paper asks a different question: how do you train recurrent networks so that neurons within the network produce their own target function? Here the target can be either firing rate or synaptic drive. What's interesting about this paper is that the authors give the conditions under which training is possible. These rules provide insight into the limits of performance of spiking RNNs, something that has, I believe, been lacking in the field.

Essential revisions:

Reviewers A and B's full reviews, along with the subsequent discussion are included below. You should focus on making A happy (which is pretty easy, since he/she had only minor comments). However, when you revise the paper, you probably should keep B's review in mind – his/her interpretation of your paper may be shared by others.

*Reviewer A comments:*

The focus of most (maybe all?) research on RNNs has been on output: how do you train recurrent networks to produce a low dimensional (compared to the dimensionality of the network) target function? This paper asks a different question: how do you train recurrent networks so that neurons within the network produce their own target function? Here the target can be either firing rate or synaptic drive.

What's interesting about this paper is that the authors give the conditions under which training is possible:

- The target functions have to be sufficiently diverse..

- A sufficiently large number of neurons have to be associated with targets.

- The synaptic time constant can't be too long compared to the timescale of the target functions.

- the synaptic time constant can't be too short compared to 1/N (N is the number of neurons in the network).

- The training period can't be too long compared to N.

These rules provide insight into the limits of performance of spiking RNNs, something that has, I believe, been lacking in the field. [Minor comments not shown.]

*Reviewer B comments:*

The authors present an interesting study examining the capability of recurrent networks of QIF neurons for learning various types of signals. They apply a RLS learning rule so that the network produces various periodic functions, filtered noise, the activity of chaotic networks, and experimental data. They present evidence to support their contention that two key conditions need be met for learning: (1) including a sufficiently rich set of target functions; (2) appropriately matched synaptic time-scale relative to the target dynamics. Additionally, a mean field analysis was performed and shown to be consistent with dynamics after learning.

While a handful of curious findings are presented, for the following two reasons I do not believe the work, as presented, is suitable for publication in this journal.

First, the examples, learning conditions and supporting experiments do not constitute a significant advance over previous work. While I cannot refer to prior work that has studied the ability to produce the specific signals presented, the authors do not properly motivate the importance of these particular signals. Prior publications that train RNNs (spiking too) typically focus on constructing networks that can perform tasks, where the motivation for the signals they construct is obvious: they use signals necessary for performing computations. For three of the examples (rate chaos, OU noise and spiking chaos) it's not clear what computation these networks might perform after learning (in fact, the primary motivation for stabilizing rate chaos, e.g. in FORCE learning, was so that the network COULD produce a computation after learning). In the case of the work of Laje and Buonomano (which the authors claim to extend to spiking networks), they sought to stabilize chaotic rate trajectories as a model for understanding timing tasks; here the authors do not present any use for these dynamical patterns, just that they can be learned.

Second, the presented work is simultaneously too technical (and therefore too specific) and too vague to be of significant interest to the general neuroscience community. The mean field analysis is interesting and stands out when compared to other published works, which have not extensively studied how analytic solutions compare to simulations in trained networks. Nevertheless, this is a minor, technical point, not likely to be interesting to most neuroscientists (unless the authors can explain why this is interesting).

While the question of universal computation is of general interest, the presented results are not sufficiently rigorous to match the claims. While they claim that previous work has been impressive but not sufficient to showcase the full richness of what can be expressed by spiking RNNs, the authors claim to learn "arbitrary" and "near universal" dynamics but only show examples (as previous work has) from a rather small set of potential dynamics. If this paper were to be published, the authors would need to clarify what they mean by this to accurately reflect the presented results or include a proof of the universality they are claiming to show. Another question of general interest is what constitutes a "sufficient basis" for learning but here again the authors only provide a few examples that are not shown to extend beyond the direct application studied. This approach is inferior to previous methods, since no prescription is offered for establishing sufficiency other than generating additional random signals; in previous work where specific computations were examined, the only signals required were those necessary for performing the computation and these could be "bootstrapped" by exploiting knowledge about the train-ability of firing rate networks. The authors offer a bold claim (identifying the sufficiency of a rich basis) with pretty weak support (just keep adding random signals).

I conclude by highlighting the most interesting and promising result: training with neural data. That the training required auxiliary dynamics is interesting and provides a prediction of a set of dynamical patterns that should exist in the brain in order for the network to perform the task it did. Expounding on these results would yield a very interesting finding, albeit beyond the scope of a simple revision to the present work.

*A's response to B's comments (during discussion phase):*

My main feeling is that they are looking at a very different problem than anybody else: they're training every single neuron (or a large fraction of neurons) to produce target functions, with different target functions (but similar timescales) for each neuron. This is, in my opinion, completely unrealistic – it's much more realistic to train networks to produce relatively low dimensional output patterns. But what I thought was interesting was that the paper gave bounds on performance in general in terms of timescales of the network and the target output of each neuron. If you can't produce target functions with a particular timescale when you can train each neuron individually, I would think that you certainly can't train the network as a whole to produce target functions with that timescale. Which is a point they might have emphasized more, but that would be easy to fix. But maybe I'm interested simply because it was something I always wondered about.

With this view, your first objection, which is that they didn't consider interesting functions, becomes, I think, moot. Your second objection is that the work is too technical and too vague. I agree the mean field analysis is very technical, and will be followed by only a handful of people. But it doesn't seem to be an integral part of the paper. And I think the rest is pretty easy to follow. Not sure what you meant by too vague – could you expand?

You also say that their results are not sufficiently rigorous. However, if they can make the arguments rigorous, would you be satisfied, at least about the rigorousness (but not necessarily about the first two points)?

*B's response to A:*

I disagree with you that they are looking at a different problem than anybody else. However, none of this works addresses the conditions for learning, I agree with you there.

I agree with you that understanding the limits of learning in RNNs is interesting and lacking, but to me, their results related to that point only account for a fraction of the full paper. As written, that point is embedded within a range of other findings that are unmotivated (e.g. the various other signals they train networks to perform that do not support their results about the limits of learning), incomplete (using data to train networks, as I wrote in my review) and that do not ultimately support the point about learnability. However, if they were to remove these other results, the point on learnability alone would not constitute a full paper.

In my view, they introduce two main conditions for learning: that the synaptic timescale should roughly match the targets and that the targets should be "suffciently rich". The second condition highlights my concerns about the vagueness and rigorness of the results. They illustrate the need for sufficient richness by removing targets from some of the training sets they used and show that these network cannot learn (Figure 3). While this is anecdotally interesting, I do not believe that illustrating that this is true in a handful of cases is sufficient to say anything general about the conditions necessary for learning. At best, this is good enough to make some pretty hand-waving arguments about the need for an overcomplete basis, which is precisely what they do in Section 2. Again, it's not that I don't find this interesting, I just don't seem much substance there.

Their first condition for learning is definitely much more rigorously addressed (the simulations and calculations of Figure 6 are nice) but I find this neither surprising or groundbreaking (you can't learn slow signals without slow synapses, or fast signals without fast synapses....) I think this point could be stated much more simply without the need for the various training targets they used (chaotic rate models and stochastic noise) which again, strengthens my resolve that much of what they do feels unmotivated. Even though Figure 6 being nicely done, 1 figure is not enough for a paper.

Finally, I think the tasks they are learning are perfectly "interesting", just unmotivated.

---

## [Author Response]

We would like to thank the reviewers for their valuable comments. We have made the code for learning recurrent spiking dynamics available on-line at https://github.com/chrismkkim/SpikeLearning

We will use following summary as reference points when responding to the reviewers’ comments:

S1) As Reviewer A pointed out, most research on reservoir computing in RNNs has focused on training the network output to perform a specific computation [Sussillo and Abbott, 2009; Laje and Buonomano, 2013; Rajan, Harvey, and Tank, 2016; DePasquale et al., 2018]. The implicit idea is that the RNN’s job is to provide a reservoir of trajectories that can be combined linearly with trainable weights to produce a desired output. In contrast, we ask the question of whether the output neurons are necessary at all. What if the RNN could be trained such that the output and reservoir were one and the same? And can this be done in a spiking network? None of these questions had answers before our work.

S2) In order for the reservoir computing scheme to work the RNN needs to be rich enough and repeatable, which is nontrivial to attain [Sussillo and Abbott, 2009] achieves this by stabilizing an inherently chaotic network with feedback or training, [Laje and Buonomano, 2013] trains the weights of the RNN to stabilize specific chaotic trajectories, [Rajan, Harvey, and Tank, 2016] used sequential activity derived from experimental data, and [DePasquale et al., 2018] used network activity extracted from a “target network”.

S3) We identified two conditions under which learning is successful for any arbitrary signal: (1) the synaptic time scales must be fast enough and the network size must be large enough and (2) the set of target functions must be sufficiently heterogeneous.

Response to essential revisions:

Reviewer B commented that the set of target functions a network learns (e.g. OU noise) was not motivated for performing computations. We respectfully disagree. As stated in (S1), our goal was to show that an RNN could produce arbitrary patterns such that there would be no need to distinguish between the RNN and the output neurons. The question we address is specifically the computational capability of an RNN in terms of the patterns it can produce. As stated in (S2), previous studies considered only specific forms of target functions for the RNN, hence it remains unknown whether a spiking network can learn “arbitrary” functions. Laje and Buonomano even stated (see their Online Methods) that “choosing innate trajectories from another network did not produce effective training”, alluding that recurrent networks may not be able to learn arbitrary functions. Our examples show that this is not the case and that individual neurons may be able to perform universal computations. The target functions considered in Figure 3 (periodic functions, rate chaos, OU noise) provide a few examples of random functions that a network can learn. We felt that this provided a generic selection of arbitrary functions, and also showed that the network is able to reproduce firing patterns of an animal performing a task. We revised the introduction of the original manuscript and added additional explanation in the first paragraph of discussion to make it clear that our goal was to investigate the computational capability of RNNs to encode recurrent spiking dynamics.

We also wish to correct Reviewer B’s interpretation of the first condition for learning (small sampling and tracking errors) since his statement “you can’t learn slow signals without slow synapses, or fast signals without fast synapses” is not correct and we apologize for not making this point more clear in our original text. It is possible to learn slow signals without slow synapses. As stated in (S3), the first condition requires that synapses to be fast enough (i.e. small tracking error,etrack~ts/tc) and network size to be large enough (i.e. small sampling error,esample~1/tsN), then the network can learn any signals, in particular the slow ones. As long as the synapse is faster than the target (i.e. their time scales need not be comparable), and there are plenty of neurons in the network to produce continuous target signals, forming appropriate recurrent connections can compensate for the short synaptic time scale and allows the network to generate slow signals. This is the main conclusion of the error analysis and is stated immediately after Equation 3. Our simulation results also support the error analysis: in Figure 6A, the parameter regime above the diagonal line (i.e. slow signal and fast synapses, *t_c_ > t_s_*) learns successfully, and similarly learning is successful if *t_s_/t_c_ <* 1 in Figure 6E. Such slow dynamics, consistent with our results, have been studied in random rate networks [Sompolinsky, Crisanti and Sommers, 1988], random spiking networks [Ostojic, 2014] and random rate networks when reciprocal connections are over-represented [Mart´ı, Brunel and Ostojic, 2018]. The example shown in Figure 6B also demonstrates that networks with fast synapses (*τ_s_
*= 30ms) can learn slow signals (*τ_c_
*= 100ms). We pointed out these facts in the third and fourth paragraphs of Results section. What may have led to the confusion is that for a fixed network size, synapses cannot be too fast or too slow, as there is a trade off between tracking and sampling error (See inverted “U”-curves in Figure 6C,D). However, as we also show, the sampling error can be overcome with more neurons so synapses can be arbitrarily fast as long as the network can be arbitrarily large (Figure 6D).

The second condition for learning is that the set of target functions need to be sufficiently heterogeneous. Reviewer B criticized that our definition of the second condition is vague and lacks rigor since we only provided anecdotal examples in Figure 3. However, we actually discussed the mathematical basis for the second condition in subsection “Learning capacity increases with network size” in the original manuscript. We now realize that our original treatment was not convincing and have strived to improve it so that it better articulates our argument. Here we present the main ideas of the mathematical argument (See the revised manuscript for details).

The network dynamics under the quasi-static condition can be expressed in a matrix form *U* = *WV* where *U* and *V* ≡ *φ(U* + *I*) are *N* × *P* matrices. Here, the *i*^th^ row of *U* and *V* is the time discretized trajectory of neuron *i*’s synaptic drive and spiking rate, respectively. The question of successful learning is reduced to analyzing the solution set of *W* to this system of equations. In principle, learning is possible when the rows of *U* are spanned by the rows of *V,* i.e. target functions can be self-consistently generated by firing rate patterns induced by the targets. We define target functions to be sufficiently heterogeneous if the firing rate matrix *V* is full-rank because under such condition that the image of *V* encompasses the largest possible space.

We show that rank(*V*) being maximal is sufficient for an exact or approximate solution *W* to exist. When *N* ≥ *P*, the maximal rank(*V*) implies that the rows of *V* span the entire Euclidean space R*^P^*, of which the target functions (i.e. rows of *U*) are elements. Equivalently, we can say that the number of unknowns (*N*) exceeds or equals the number of independent equations (*P*). Therefore, multiple perfect solutions are possible, and the regularization term is required for the learning scheme to converge to one solution. When *N < P*, the maximal rank(*V*) implies that the linearly independent rows of *V* span an *N*-dimensional subspace of R*^P^*. The number of unknowns (*N*) is less than the number of equations (*P*), so perfect learning is not possible and we can only find an approximate regression solution *W* = *UV^T^(VV^T^*)^−1^ where the inverse of *VV^T^
*exists because rank(*V*) is maximal. If rank(*V*) is not maximal, a solution can still exists if rows of *U* are close to the subspace spanned by *V.* However, in this case the success depend on the specific choice of target functions because the dimension of the subspace spanned by *V* is strictly less than *P*, so whether the rows of *U* are contained in or close to this subspace is determined by the geometry of the subspace. This shows why increasing pattern heterogeneity, which makes the rows of *V* independent and the rank higher, is beneficial for learning. Conversely, if a large number of neurons is trained on the same target, it becomes increasingly difficult to develop the target patterns with limited basis functions. Given that this mathematical explanation is easily missed, we discussed the underlying mathematics in detail after introducing the second condition for learning in the fourth paragraph of Results section in the revised manuscript.